



**What controls fire size in the South American Gran Chaco?**
**Exploring atmospheric, landscape, and anthropogenic drivers.**
Rodrigo San Martín[1], Catherine Ottlé[1], Anna Sorenssön[2,3,4], Pradeebane Vaittinada Ayar[1], Florent
Mouillot[5], Marielle Malfante[6]
[1]Laboratoire des Sciences du Climat et de l'Environnement, LSCE/IPSL, CEA-CNRS-UVSQ, Université Paris-Saclay, Gif-
sur-Yvette, France;
[2]Centro de Investigaciones del Mar y la Atmósfera (CIMA), CONICET – Universidad de Buenos Aires, Buenos Aires,
Argentina;
[3]CNRS, CNRS – IRD – CONICET – UBA, Instituto Franco-Argentino para el Estudio del Clima y sus Impactos (IRL 3351
IFAECI), Argentina;
[4]Facultad de Ciencias Exactas y Naturales, Universidad de Buenos Aires, Buenos Aires, Argentina;
[5]UMR CEFE, University of Montpellier, CNRS, EPHE, IRD, Montpellier, France;
[6]Univ. Grenoble Alpes, CEA, List, Grenoble, France;
*Correspondence to*: Rodrigo San Martin (rodrigo.sanmartin@lsce.ipsl.fr)



**Abstract.** Wildfires are key ecological agents in the Gran Chaco, one of the world's largest dry forest ecosystems, where fire regimes are increasingly shaped by human pressure and climate variability. However, the drivers of fire size variability remain poorly understood. We analysed over 100,000 fire patches (2001–2022) from the FRYv2.0 database to assess environmental controls on fire size and morphology across the Wet, Dry, and Very Dry Chaco. High-resolution fire polygon data were combined with ERA5-Land reanalysis, vegetation and topographic metrics, and anthropogenic layers. Fire sizes were highly skewed: >80% were <5 km², yet large events (Megafires >100 km², Gigafires >1000 km²) dominated burned area (BA). Gigafires were rare but mostly confined to the Dry Chaco, whereas the Wet Chaco had the highest BA, fire frequency, and Megafire count. Fire Weather Index (FWI)–BA correlations reached r = 0.7 in the Wet Chaco but were weaker and spatially fragmented in drier subregions, where fuel continuity and ignition context played larger roles. Lag analyses showed that in drier areas, wet-season biomass buildup (4–6 months prior) increased subsequent fire activity, while in wetter areas short-term dryness (1–3 months prior) was more predictive. During-fire meteorology, especially persistent strong winds, better explained fire morphology than pre-fire conditions. Random Forest models ranked static landscape features (elevation, land-cover evenness, slope, tree cover) highest in size prediction. Our results reveal region-specific fire–environment couplings, clarifying the interplay of meteorological, ecological, and anthropogenic factors, and providing actionable insights for fire risk forecasting and management in the Gran Chaco.





## 1 INTRODUCTION

Wildfires shape global ecosystems by influencing vegetation structure, biodiversity, and landscape
composition (Bowman et al., 2009; Archibald et al., 2013; Chuvieco et al., 2020). The Gran Chaco,
spanning parts of Argentina, Bolivia, Paraguay, and Brazil, is one of the largest remaining dry forest
ecosystems, with marked variation in precipitation, vegetation, and human land use (Morello &
Adámoli, 1968; Olson et al., 2001; Ginzburg et al., 2005; Torrella & Adámoli, 2005). Fire has long
modulated its forest structure and driven transitions between forests, shrublands, and grasslands
(Bucher, 1982; Kunst et al., 2003; Vidal-Riveros et al., 2023).
In recent decades, Gran Chaco fire regimes have shifted under land-use intensification and climate
variability (Gasparri et al., 2008; De Marzo et al., 2021; Baumann et al., 2022; Marengo et al., 2022;
Vidal-Riveros et al., 2023; San Martín et al., 2023; San Martín, 2024). These changes often produce
larger, more intense fires, especially in areas with non-native grasses or monocultures (D'Antonio &
Vitousek, 1992; Bravo et al., 2014; Vidal-Riveros et al., 2023). Natural fire breaks (e.g., water bodies)
and traditional management can limit spread (Kunst et al., 2003; Bowman et al., 2011; Archibald et al.,
2013; Bravo et al., 2014; Andela et al., 2017, 2019), while landscape heterogeneity further constrains
propagation (Bowring et al., 2024), challenging assumptions of uniform anthropogenic effects (Bistinas
et al., 2014; Archibald et al., 2018; Kelley et al., 2019). At broader scales, climatic variability—
especially rainfall patterns and drought—can outweigh land use in shaping fire size and frequency
(Krawchuk et al., 2009; Jolly et al., 2015; Jones et al., 2022).
The complexity of fire size drivers in the Gran Chaco is increasingly recognized, yet key mechanisms
remain poorly understood (Kelley et al., 2019; Jones et al., 2022; Vidal-Riveros et al., 2023, 2024).
Prolonged droughts reduce fuel moisture, increasing flammability and enabling extreme events (Alencar
et al., 2015; Naumann et al., 2023). Several major droughts coincided with strong negative El Niño–
Southern Oscillation (ENSO) phases, including the record-breaking 2020–2023 La Niña (Doblas-Reyes
et al., 2021; De Marzo et al., 2023; Meteorological Organization et al., 2023; Arias et al., 2024).
Although recent studies have advanced understanding of Gran Chaco fire regimes, key links between
patterns and meteorological or anthropogenic drivers remain unclear. Land cover and socio-
environmental factors play a major role: Baumann et al. (2022) found that deforestation pathways vary
by actor and context, influencing fire–landscape interactions; San Martín et al. (2023) showed that
precipitation–burned area (BA) relationships differ by land cover; and Levers et al. (2024) projected that
agribusiness expansion could intensify fire impacts in ecologically and socially sensitive areas.
Fire classification efforts also overlook important drivers. Vidal-Riveros et al. (2024) grouped
Paraguayan Chaco fire regimes by severity, frequency, and extent, while Naval-Fernández et al. (2025)
applied multivariate clustering of landscape attributes to delineate pyroregions in the Argentinian Chaco.



Both captured spatial variability in fire activity, but neither incorporated meteorological conditions,
limiting insights into atmospheric controls on fire behavior and size.
Research has further addressed post-fire vegetation recovery and cultural dimensions of fire. Saucedo
and Kurtz (2025) reported rapid regrowth after the 2022 megafires, followed by climate-constrained
stabilization. Sugiyama et al. (2025) highlighted Indigenous fire narratives as valuable sources of local
knowledge on ignition, spread, and ecosystem recovery.
However, no study has yet combined high-resolution meteorological data, fire morphology, and
landscape context to assess how fire size responds to both short-term anomalies and long-term
environmental patterns in the Gran Chaco.
Advances in satellite Earth observation now make this integration possible. Global BA products such as
FireCCI51 provide consistent daily burned surface estimates at moderate spatial resolutions (Chuvieco
et al., 2020). Event-based datasets like FRY (Laurent et al., 2018; Chen, 2025) and the Global Fire Atlas
(Andela et al., 2019) reconstruct individual fires from these burned pixels, enabling analysis of attributes
such as ignition date, duration, size, and morphology (Moreno et al., 2021; García et al., 2022a; Takacs
et al., 2021). In this study, we used FRYv2.0, which integrates the FRYv1.0 pixel aggregation method
with FireCCI51 BA mapping (Lizundia-Loiola et al., 2020), and combined it with environmental and
climate products to address gaps in understanding BA dynamics and fire size variability in the Gran
Chaco.
Specifically, we aim to answer the following scientific questions:
(1) What are the primary fire size characteristics and frequency in the Gran Chaco between 2001 and
2022? (2) To what extent do meteorological conditions influence the size and expansion of these fires?
(3) Beyond weather, what roles do vegetation type, topography, and human activity play in shaping fire
size and fire occurrence across the region? (4) Which of these drivers best explains the spatial and
temporal variability in fire size across the different Gran Chaco subregions?
This study adds value by providing a spatially explicit, multiscale analysis of BA and individual fire
events, clarifying fire size dynamics across landscapes from wet to arid ecosystems. By quantifying the
relative contributions of climate, landscape, and human factors, it advances understanding of fire
regimes in one of the world's most dynamic yet understudied deforestation and fire frontiers
(Kuemmerle et al., 2017; Baumann et al., 2022; Vidal-Riveros et al., 2023; Levers et al., 2024; San
Martín, 2024).



## 2 METHODS

### 2.1. Study area

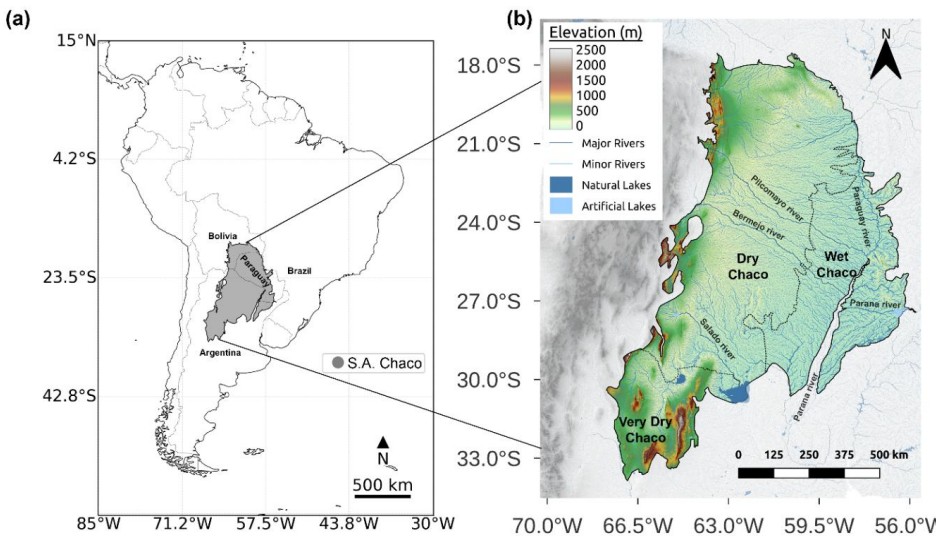

**Fig. 1.** The Gran Chaco location in South America (a) and its topography (b) with its different subregions, main rivers, and lakes. Based on Shuttle Radar Topography Mission (SRTM) at 90m (SRTM | Earthdata, 2024) and HydroSHEDS (Lehner et al., 2008).

The Gran Chaco is an extensive tropical and subtropical region of South America, covering approximately 1,100,000 km² (**Fig. 1**). It contains the world's largest continuous dry tropical forest and extensive wetland systems (Bucher, 1982; Olson et al., 2001). Terminology varies in the literature (South American Chaco, Gran Chaco, Chaco); here we use Gran Chaco for clarity.

The region is mostly flat (<200 m a.s.l.), with higher terrain in the northeast (to 500 m), Sierras de Córdoba (to 2,900 m), and Andean foothills (~2,000 m). Following Olson et al. (2001), we distinguish a humid eastern Wet Chaco from a drier western Dry Chaco, shaped by west–east gradients in precipitation, vegetation, and hydrology (Bucher, 1982; Ginzburg et al., 2005; Morello and Adámoli, 1968; Torrella and Adámoli, 2005). The Wet Chaco receives up to 1,800 mm/year and supports wetlands and palm savannas, while the Dry Chaco gets 300–800 mm/year and is dominated by drought-adapted forests. To refine this scheme, we follow Baumann et al. (2018) and designate a Very Dry Chaco in the southwest (Mendoza, San Luis, Córdoba, San Juan, La Rioja), characterized by lower biomass, greater aridity, higher elevations, and distinct fire regimes.

The Gran Chaco forms part of the La Plata basin (Musser, 2024). Rivers such as the Pilcomayo, Bermejo, and Salado originate in the Andes, cross the Dry Chaco, and disperse into megafans, streams, and wetlands in the eastern Wet Chaco. This west–east hydrological gradient drives seasonal contrasts: in



dry months, the Dry Chaco faces water scarcity, whereas the Wet Chaco retains permanent wetlands
that sustain ecological processes and fauna (Naumann et al., 2023).
The region harbors exceptional biodiversity, with over 3,400 plant species and hundreds of vertebrates,
many endemic (Redford et al., 1990; Bucher and Huszar, 1999; Nori et al., 2016).

**2.2 Datasets**


*2.2.1 Fire patches*


In this study, we used FRYv2.0, a comprehensive global database dedicated to the functional traits
(morphology, fire spread, and timing) of fire patches (FPs), to investigate fire dynamics and their
underlying drivers in the Gran Chaco. FRYv2.0 incorporates burned area (BA) data from the FireCCI51
dataset as well as from MODIS MCD64A1 in two different versions, with different temporal cut-offs of
6, 12, or 24 days, as described in Laurent et al. (2018). It offers medium-resolution FPs covering the
period from 2001 to 2022, including metrics for FPs, such as morphological traits (e.g., area, shape
index), temporal traits (e.g., burn dates, duration), dynamic traits (e.g., rate of spread, fire radiative
power, and burn severity), and land cover.
For this work, we selected the FRYv2.0 dataset based on FireCCI51 over the MODIS MCD64A1
version, due to the higher spatial resolution of the FireCCI51 input data (250 m compared to 500 m), its
suitability for the heterogeneous Chaco landscapes, and its consistency with our previous FireCCI51-
based analysis (San Martín et al., 2023), avoiding uncertainties from mixing datasets. The dataset is
available at https://osf.io/rjvz5/files/osfstorage (last accessed on 10 June 2025).

*2.2.2 Meteorological Data*


To study meteorological and climate time series in the region, we used the ERA5-Land global reanalysis
dataset focused on land surface variables, developed by the European Centre for Medium-Range
Weather Forecasts (ECMWF) (Muñoz-Sabater et al., 2021). It provides high-resolution data for land–
atmosphere interactions, designed to improve the ERA5 dataset by offering finer detail (0.1° instead of
0.25° spatial resolution) for variables affecting the land surface.
The product is available in the Copernicus Data Store (CDS) in NetCDF at
https://cds.climate.copernicus.eu/cdsapp#!/dataset/reanalysis-era5-land (last accessed on 30 May 2024).
We downloaded hourly data arrays covering January 2001 through January 2023.



*2.2.3 Environmental and Anthropogenic Data*
We compiled multiple spatial datasets to represent landscape and human-related drivers of fire activity.
Topography was derived from the Shuttle Radar Topography Mission (SRTM) digital elevation model
at 30 m resolution (https://srtm.csi.cgiar.org, accessed 26 May 2025) and resampled to 0.01° (~1 km).
Slope was calculated from the elevation surface using standard GIS tools.
Land cover (LC) was obtained from the ESA Climate Change Initiative Moderate Resolution Land
Cover (ESA CCI MRLC) product (https://cds.climate.copernicus.eu/datasets/satellite-land-cover,
accessed 26 May 2025), reclassified into groups relevant to the Gran Chaco (e.g., forests, shrublands,
grasslands, seasonally flooded herbaceous vegetation) for 2001–2022.
Human pressure variables included population density from the Gridded Population of the World v4
(CIESIN, 2017; https://www.earthdata.nasa.gov/data/projects/gpw, accessed 26 May 2025) and road
density from OpenStreetMap networks (https://www.openstreetmap.org, accessed 26 May 2025)
calculated via kernel density estimation.
Livestock    density    came    from    the    Gridded    Livestock    of    the    World    v4
(https://dataverse.harvard.edu/dataverse/glw_4, accessed 26 May 2025), resampled to match the
analytical resolution.
Soil properties (bulk density, sand content, and organic carbon at 0–5 cm depth) were obtained from
SoilGrids250m (Hengl et al., 2017; https://soilgrids.org, accessed 26 May 2026).
*2.2.3 Climate Oscillations*
To account for the influence of large-scale climate variability, we included the Multivariate El Niño–
Southern Oscillation (ENSO) Index version 2 (MEI.v2), developed by NOAA's Physical Sciences
Laboratory. The MEI.v2 time series was obtained from NOAA PSL at https://psl.noaa.gov/enso/mei/
(last accessed 26 May 2025).

**2.3 Data processing and analysis methods**
*2.3.1 Fire Weather Index (FWI)*
We built an ERA5-Land-based Canadian Fire Weather Index (FWI; Van Wagner, 1987) dataset for the
Gran Chaco at 0.1° resolution and daily time steps. We converted hourly accumulated precipitation to
hourly rainfall by differencing successive steps and summed totals from 15 UTC (day D-1) to 15 UTC
(day D), matching the FWI daily window and corresponding to local noon. We applied this fixed 15
UTC cutoff to avoid inconsistencies from varying national time zones and daylight-saving changes.
We extracted daily meteorological inputs—air temperature, relative humidity, wind speed at local noon,
and 24-h precipitation—to compute the six FWI sub-indices: Fine Fuel Moisture Code (FFMC), Duff
Moisture Code (DMC), Drought Code (DC), Initial Spread Index (ISI), Build-Up Index (BUI), and FWI.



We performed calculations with an adapted version of the FireDanger Python package
(https://github.com/steidani/FireDanger) compatible with xarray and netCDF, including pixel-level day
length for DMC and hemisphere-specific drying factors for DC.
We initialized the system on 1 January 1981 using Copernicus ERA5–FWI moisture codes at 0.25°
(Vitolo et al., 2020) interpolated to 0.1°. For anomaly analysis, we restricted the time series to 2001–
2022 to match satellite-based burned area (BA) records and calculated daily climatologies for all
variables and indices using 2001–2020 as the baseline.
*2.3.2 Fire size classification*
To better characterize fire activity across the Chaco, we classified all fire polygons (FPs) from FRYv2.0
into six size categories, ranging from very small fires (<1 km²) to gigafires (>1000 km²), following and
adapting the typology proposed by Linley et al. (2022). We used this to assess both the frequency and
relative contribution of different fire sizes across regions and seasons.

*2.3.3 Gridded burned area*
To enable a spatio-temporal comparison between fire activity from FRYv2.0 polygons and meteorology,
we developed a pipeline to transform the FP-based data into a monthly gridded product at 0.1°, matching
the ERA5-Land grid  (**Fig. 2**).

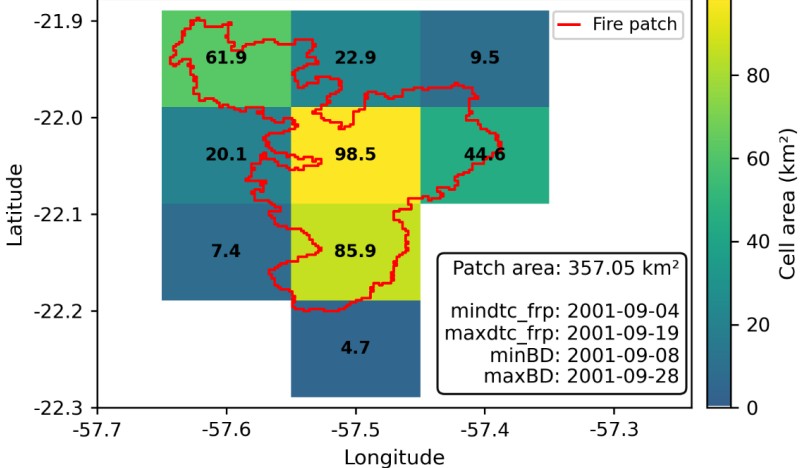


**Fig. 2.** Example of a FRY polygon (red line) over the gridded FRY dataset. Each grid cell at 0.1º is assigned the burned area corresponding to
the total fraction of the polygon that overlaps it. The values printed over each grid cell correspond to these values.




The temporal assignment of fires to months followed a hybrid strategy: where MODIS-derived hotspot
detection dates (*mindtc_frp* and *maxdtc_frp*) were available in a given FP (typically absent in very small
FPs) they were used. Both FireCCI51- and MODIS-based versions of FRYv2.0 include these hotspot
date variables when available for the FP. When hotspot dates were missing, we used the FireCCI51-
derived burn dates (*minBD* and *maxBD*), which are based on surface reflectance changes and are
available for all FPs. For FPs spanning multiple months, we assigned the fire to the month in which it
started, unless its duration in a subsequent month exceeded that of the starting month by more than two
days.
Each FP was rasterized over the ERA5 grid by intersecting it with individual cells. The intersected area
in square kilometers was computed using the WGS84 ellipsoid model. These contributions were
aggregated per cell and per assigned month to build a three-dimensional array of monthly BA (*lat x lon*
*x time*). A similar procedure was implemented for fire counts, using ignition coordinates when available.
Each FP's fire ignition coordinate was allocated to the closest cell in the 0.1º grid. The resulting monthly
gridded dataset included two variables: BA and counts.

*2.3.4 Fire-weather types*
We classified fire patches (FPs) into three groups based on associated atmospheric conditions using the
K-means clustering algorithm (MacQueen, 1967) in scikit-learn v1.3. This approach follows prior
applications in fire studies (Ruffault et al., 2016, 2020; Vidal-Riveros et al., 2024) and aimed to identify
distinct fire-weather types and assess their influence on fire size and shape.
We retained only FPs between 1 and 100 km² (N = 76,263) to reduce biases from very small or very
large events. For each FP, we extracted daily ERA5-Land meteorological data and generated FWI time
series from 7 months before ignition to 7 months after. Two feature sets were built: one for pre-fire
conditions and one for during-fire conditions.
For the *Pre-Fire* set, we used normalized anomalies of 2-m air temperature, 10-m wind speed, relative
humidity (RH), drought code (DC), and duff moisture code (DMC) (Ruffault et al., 2020). Pre-fire
values were calculated as the 3-day mean from ignition day (D) to D-2 to limit detection-date bias
(Lizundia Loiola et al., 2020; Pettinari et al., 2021) while avoiding noise from longer lags.
For the *During-Fire* set, we computed the same variables averaged over the fire's duration and added a
metric specifically designed to capture the role of strong, persistent winds in shaping fire behavior: the
Extreme Wind Directionality Index (*EW_dir_index*). This index measures both how often extreme
winds occurred and how steady their direction was.
The first component, fraction of extreme-wind days (*EW_frac*), is the proportion of burning days when
the daily maximum wind speed exceeded 25 km h⁻¹:
(**Eq. 1**):



$$\text{EW\_frac} = \frac{EW}{N}$$
where $EW$ is the number of days with extreme winds and $N$ is the total fire duration (days).
High values indicate that strong winds occurred on many burning days.
The second component, wind direction steadiness (*wind_dir_R*), reflects how consistent the wind
direction was across the fire's duration (N). Each day's mean wind direction ($\theta_i$, in radians) is
represented as a unit vector, summed across all days, and normalized by the fire duration:
(**Eq. 2**):
$$\text{wind\_dir\_R} = \frac{\sqrt{\left(\sum_{i=1}^{N} \cos \theta_i\right)^2 + \left(\sum_{i=1}^{N} \sin \theta_i\right)^2}}{N}$$
Values near 1 mean winds blew in a stable direction throughout the event, while values near 0 mean
wind directions shifted substantially from day to day.
The *EW_dir_index* is the product of *EW_frac* and *wind_dir_R*:
(**Eq. 3**):
$$EW\_dir\_index = EW\_frac \times wind\_dir\_R$$
It reaches high values only when strong winds occur on many burning days and blow consistently from
the same direction, identifying fires likely driven by sustained, unidirectional wind conditions.
All variables were standardized (mean = 0, $\sigma$ = 1) before clustering. The resulting data matrix (nnn fires
$\times$ ppp variables) was clustered with k = 3, squared Euclidean distance, k-means++ initialization, 50
random restarts, and a convergence tolerance of $10^{-4}$. We retained three clusters based on a prior
hypothesis (wind-driven, drought-driven, and neutral), an elbow in the within-cluster sum-of-squares
curve, and a peak in the silhouette coefficient at k = 3.
Cluster labels were assigned by interpreting centroid positions in principal component space and
examining the temporal evolution of variables (**Fig. A1**). Robustness was assessed using mean silhouette
coefficients and their distribution across clusters. The first two principal components explained more
than 60 % of the variance and clearly separated cluster centroids.

*2.3.5 Fire size drivers*
To investigate the role of environmental and anthropogenic variables in shaping fire activity, we
extracted a diverse set of FP-level predictors encompassing topographic, climatic, anthropogenic,
vegetation, and landscape heterogeneity dimensions. These variables, listed in **Table 1**, were used as
inputs in the Random Forest (RF) models to assess their relative importance in explaining fire size and
frequency.





**Table 1.** Polygon-level predictor variables used in the Random Forest models, grouped by variable type.

| Category | Variables |
|---|---|
| **Topographic** | Mean Slope (%) <br> Mean Elevation (m) |
| **Climatic (during fire)** | Precipitation (mm) <br> Maximum Wind Speed (km/h) <br> Extreme Wind and Direction Index (EW_dir_index) <br> Extreme Wind Days Fraction (EW_frac) |
| **Anthropogenic** | Cattle Density (heads/km²) <br> Road Density (km/km²) <br> Population Density (p/km²) |
| **Vegetation productivity** | LAI for previous growing season (MODIS-derived) |
| **Land Cover Composition** | Flooded Herbaceous vegetation (%) <br> Tree Cover (%) <br> Shrublands (%) <br> Trees/Shrubs/Herbs Mosaics (%) <br> Natural/Croplands Herbaceous Mosaics (%) |
| **Landscape Heterogeneity** | Land Cover Diversity (Shannon Index, H) <br> Land Cover Evenness (Pielou Index, E) |


The Shannon diversity ($H$) and Pielou's evenness ($E$) were computed as follows:

(**Eq. 4**) Shannon Diversity Index (Shannon, 1948):
$$H = -\sum_{i=1}^{m} p_i \ \log(p_i)$$

Where $m$ is the number of land cover classes present in the polygon, $p_i$ is the proportion of land cover
type i, and the sum includes all classes with $p_i > 0$.

(**Eq. 5**) Pielou's evenness (Pielou, 1966):
$$E = \frac{H}{\log(m)}$$

Where $H$ is the Shannon Diversity Index and $m$ is the number of land cover classes present in the
polygon.
Once all predictor variables were derived, we trained RF models using a set of 17 explanatory variables
to analyze the drivers of fire behavior, using the variable *n_cell* from the FRY dataset as the response
variable. This variable represents the number of FireCCI51 pixels within each FP and was preferred
over polygon-based *area* due to the latter's dependency on latitude, which introduced artificial
discontinuities. In contrast, *n_cell* provided a discrete and spatially consistent proxy for BA, improving
model stability and interpretability.



We implemented 12 RF models across five configurations: (i) a global model using all 76,263 polygons
(1–100 km²); (ii) three subregion-specific models for the Wet, Dry, and Very Dry Chaco; (iii) two
seasonal models based on ignition season (wet vs dry); and (iv) two sets of three cluster-based models
(pre-fire and during-fire conditions) derived from the meteorological classification (see Section 2.3.4).
All models were trained using the *ranger* R package (Wright and Ziegler, 2017) with quantile regression
forests (Meinshausen, 2006). We used 500 trees, a minimum node size of 5, variance-based importance,
and the Poisson split rule, with 4 variables considered at each split. Feature selection included correlation
filtering ($r > 0.8$ threshold) and preliminary importance scores. Each model was trained on 75% of the
data and validated on the remaining 25%. We evaluated feature contributions using SHAP (SHapley
Additive exPlanations) values.





# 3 RESULTS

## 3.1 Burned area and ignitions

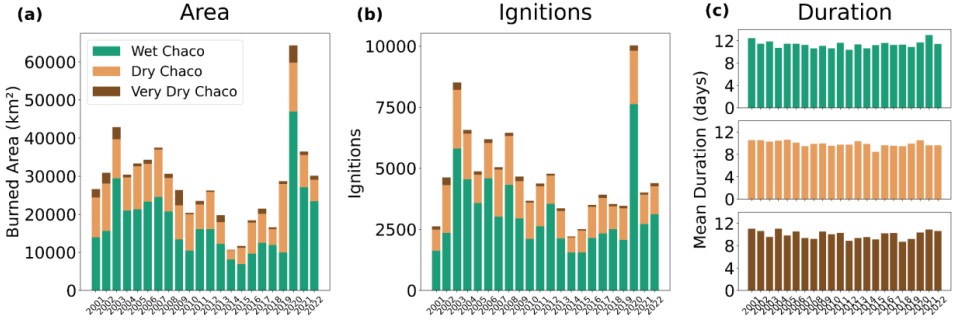

**Fig. 3.** Total annual burned areas (a), ignitions (b), and mean fire durations (c), between 2001 and 2022 in the Wet, Dry, and Very Dry Chaco regions. Extracted from FRYv2.0.

We examined the interannual relationship between total burned area (BA) and the number of fire polygons (FPs) across the Chaco (**Fig. 3**). Overall, BA and ignition counts show a positive association, though with regional and seasonal variability. In the Wet Chaco, strong correlations were found in both wet and dry seasons ($R^2$ = 0.96 and 0.91), indicating fire extent is largely proportional to ignition frequency (**Fig. A2**). The Dry Chaco also showed a high wet-season correlation ($R^2$ = 0.87), but a weaker dry-season one ($R^2$ = 0.45), suggesting a greater role of other drivers in the latter. In the Very Dry Chaco, wet-season fires were sparse and weakly correlated with BA (R = 0.11), while a stronger correlation emerged in the dry season ($R^2$ = 0.78). Mean fire duration remained relatively stable over time, implying that interannual variability in BA is primarily linked to ignition frequency and fire size, rather than duration.



**3.2 Fire size distribution and regional differences**

**Fig. 4.** (a) and (b) Land-cover distribution in the Gran Chaco based on ESA-CCI MRLC for 2001 and 2022, respectively. (c) Forest transition classes between 2001 and 2022, showing forest loss (forest to non-forest), forest gain (non-forest to forest), and stable forest. Forests include all tree cover classes; non-forest pixels appear in grey. (d) Spatial distribution of fire events (2001–2022) categorized by fire size using FRYv2.0 data. Fire-size classes range from Very Small (< 1 km²) to Gigafires (> 1000 km²). Fires polygons overlapping the Chaco boundary are retained.






**Fig. 4** shows the LC distribution of the Gran Chaco in 2001 and 2022 (panels a and b), the spatial pattern
of forest transitions between 2001 and 2022 (panel c), and all fire events recorded during 2001–2022
categorized by fire size (panel d). The Wet Chaco is dominated by seasonally flooded herbaceous
vegetation, forest mosaics, productive grasslands, and croplands, and it exhibits the highest fire
frequency. In contrast, the Dry and Very Dry Chaco regions show increasing proportions of shrublands,
fragmented forests, and agricultural frontiers.
Fire size distribution is strongly right-skewed across all subregions: over 80 % of events fall within the
Very Small (< 1 km²) and Small (1–5 km²) categories (Table A1; Fig. A3). Larger fires, although less
frequent, account for a disproportionate share of total burned area. While Very Small to Large (10–100
km²) fires are widespread, Megafires (100–1000 km²) are most common in the Wet Chaco, likely due
to continuous fuel beds in grasslands and wetlands. These large fires often occur in areas dominated by
seasonally flooded herbaceous vegetation, which can generate high flammability during dry periods.
Gigafires (> 1000 km²), although rare, are almost exclusively observed in the Dry and Very Chaco.
Forest loss is widespread across the Chaco in all three countries, with extensive deforestation frontiers
in both Argentina and Paraguay. However, the association between fires and these frontiers differs
regionally. In Argentina, deforestation zones often coincide with clusters of small and medium fires,
whereas in Paraguay and Bolivia fire activity is less evident along recent forest loss edges. In all regions,
most large fires occurred in non-forest areas. Shrublands were excluded from the forest class definition,
which here only includes tree-cover categories.




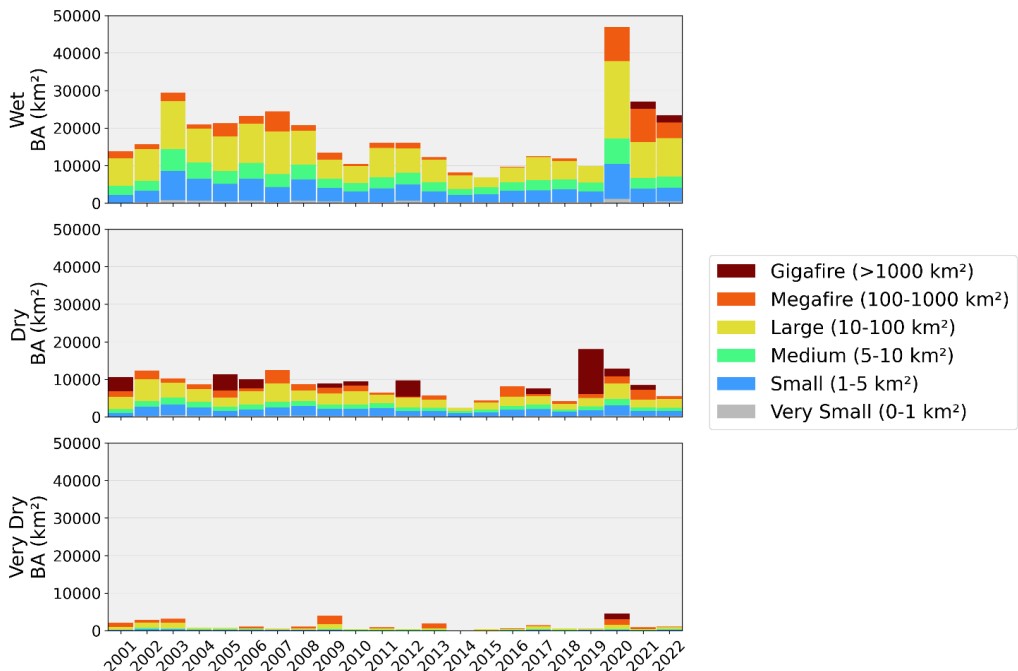

**Fig. 5.** Cumulative burned area (2001–2022) by fire-size class across the Wet, Dry, and Very Dry Chaco subregions.

According to **Fig. 5**, the Wet Chaco registers the highest total burned area, nearly double that of the Dry and Very Dry regions. In this subregion, Large fires contribute ~40% of annual BA, and Small fires ~20% (**Fig. A4**). Despite their modest size, small fires contribute substantially to BA in the Wet Chaco due to their high frequency between 2001 and 2022 (>36,000). Extreme years such as 2003 and 2020 were marked by widespread outbreaks.

In the Dry Chaco, fire frequency is lower, but large fires play a more prominent role. Large fires account for about 25% of the annual burned area, and Gigafires can dominate totals in some years. For example, in 2019, just three Gigafires in the Dry Chaco burned approximately 10,000 km², which corresponds to the region's mean annual BA and represented more than 50% of the total for that year.

The Very Dry Chaco, while recording the lowest overall BA, exhibits abrupt interannual peaks driven by isolated Megafires and Gigafires, pointing to a more stochastic fire regime.

Between 2020 and 2022, the Wet Chaco experienced an unprecedented number of Megafires and Gigafires, both in terms of event counts and their contribution to total BA. These patterns align with the extreme fire-weather anomalies described in **Section 3.3**.



### 3.3 Fire–weather relationship

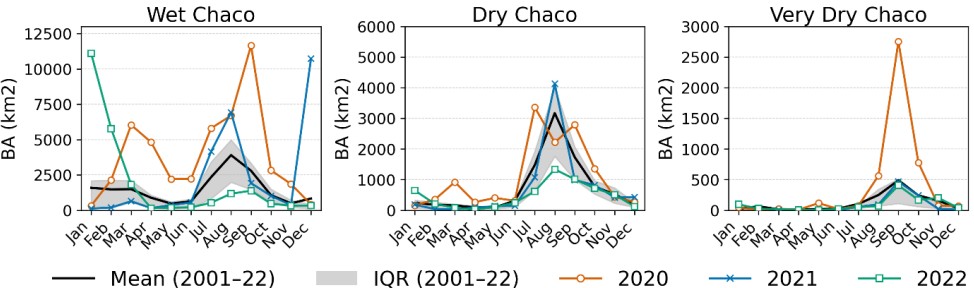

**Fig. 6**. Seasonality of burned area (BA, km²) in the Wet, Dry, and Very Dry Chaco. The black curve is the 2001–2022 monthly mean and the grey band shows the interquartile range (25–75%). Colored curves overlay monthly BA for 2020 (orange circles), 2021 (blue crosses), and 2022 (green squares), highlighting differences from the climatological envelope. Y-axis limits differ by panel.

**Fig. 6** presents the monthly BA climatology (2001–2022) with 2020–2022 overlaid for the Wet, Dry, and Very Dry Chaco. In the Wet Chaco, BA in 2020 is above average for most months, with a secondary pulse in March–April (late wet season) preceding pronounced peaks in August–September (winter/dry season). In contrast, anomalies in 2021–2022 are concentrated in the summer/wet season (December–March), reaching levels similar to the typical late-winter/early-spring maximum, while post-winter months in 2022 remain mostly below average. In the Dry Chaco, 2020 stands out as extreme, particularly in July and September, whereas 2021 records an exceptional August at or above historical maxima and 2022 stays near or below the mean. In the Very Dry Chaco, positive anomalies are dominated by 2020, with a sharp October maximum; 2021 shows only minor increases, and 2022 remains subdued. Overall, 2020 shows widespread positive anomalies lasting several months across all subregions. In contrast, 2021 and 2022 generally feature shorter peaks, often concentrated in summer, although 2021 also records exceptional winter fires in the Dry Chaco. Activity during the canonical late-winter fire season is otherwise limited, particularly in 2022.

Spatial patterns of fire–weather coupling are explored in **Fig. 7**, which shows the per-pixel Pearson correlation between monthly Fire Weather Index (FWI) anomalies and BA during wet and dry seasons. Significant positive correlations ($p < 0.05$) are concentrated in the Wet Chaco, where coefficients reach up to 0.7 during the wet season. In contrast, the Dry and Very Dry Chaco show weaker and more spatially scattered relationships, partly due to lower fire frequency.

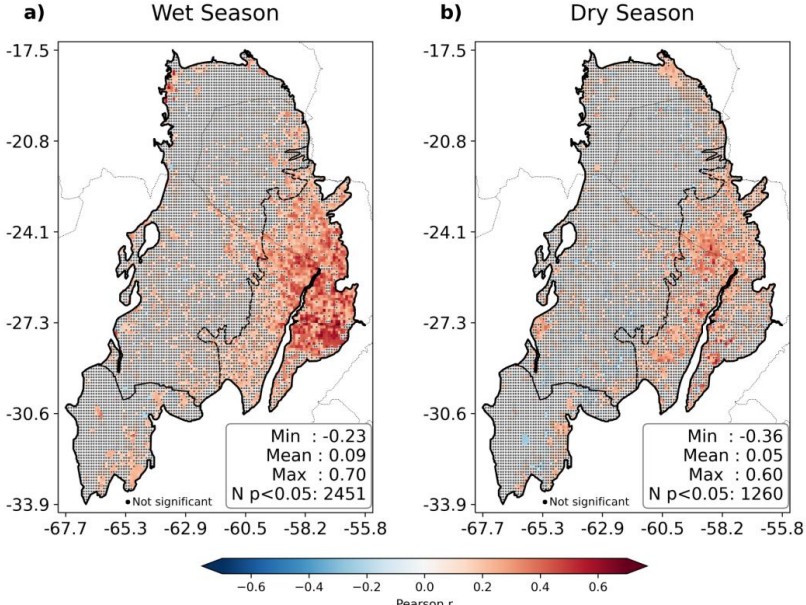

**Fig. 7.** Spatial distribution of pixel-wise Pearson correlation coefficients between monthly Fire Weather Index (FWI) anomalies and monthly burned area (BA) for the period 2001–2022: (a) Wet Season and (b) Dry Season. The color bar indicates the strength and direction of the correlation (from negative in blue to positive in red). Inset statistics summarize the distribution of coefficients (Min, Mean, Max). Pixels marked with small black circles represent non-significant correlations (p-value > 0.05), while unmarked pixels indicate significant correlations (p-value < 0.05). Only pixels with more than 3 time steps with burned area >0 were kept to avoid biased correlations related to very few or no fires.

To further explore the spatial sensitivity of fire activity to fire weather, **Fig. 8** compares per-pixel correlations between monthly FWI anomalies and two metrics: fire counts (ignitions) and BA. Each dot represents a 0.1° grid cell, and quadrants classify response types. In the Wet Chaco, 93% of cells fall in Q1, where both metrics show positive correlations with FWI, with moderate mean values ($0.17 \pm 0.12$ for ignitions, $0.19 \pm 0.13$ for BA) and strong inter-metric correlation ($r = 0.76$). The Dry and Very Dry Chaco show more heterogeneous patterns, with Q1 proportions of 59% and 61%, and weaker mean correlations (~0.04–0.06). Still, inter-metric spatial correlations remain high ($r = 0.81$ and $r = 0.72$), indicating that regions more sensitive to fire weather in terms of ignitions also tend to be more sensitive in terms of fire extent.



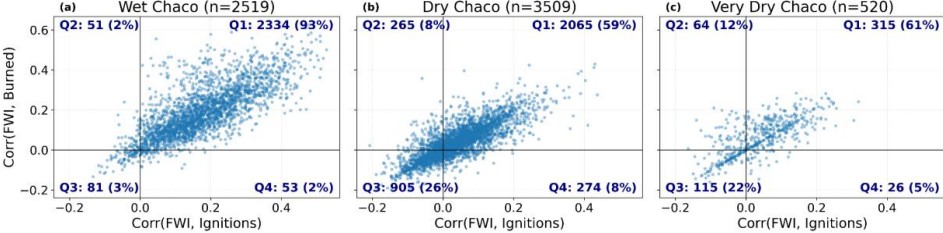

**Fig. 8.** Each panel shows a scatterplot of per-pixel Pearson correlation coefficients between the Fire Weather Index (FWI) and two fire activity metrics—ignition frequency (x-axis) and burned area (y-axis)—over the period 2001–2022. The panels correspond to the Wet, Dry, and Very Dry Chaco subregions, and each dot represents a 0.1° × 0.1° grid cell. Quadrants are defined by the sign of each correlation coefficient to classify spatial patterns of fire–weather association: Q1 (top-right) includes pixels with positive correlations for both ignitions and burned area; Q3 (bottom-left) includes negative correlations for both; Q2 and Q4 represent divergent cases. For each subregion, quadrant counts, percentages, and summary statistics (mean ± standard deviation of each correlation axis and Pearson r between them) are annotated.

Finally, the temporal co-evolution of annual BA and FWI anomalies is illustrated in the appendix (Figs. A5–A6). Several years, especially in the Wet Chaco, show strong spatial correspondence between extensive fire activity and positive FWI anomalies (e.g. 2012, 2020–2022). However, other years (e.g. 2003) reveal extensive BA without matching FWI extremes, underscoring that weather is not the sole driver of interannual variability.

### 3.4 Temporal dynamics of fire–environment interactions

To explore how conditions evolve before and after fire events, we analyzed both regional time series and lagged correlations between BA anomalies and three key drivers: FWI, rainfall, and vegetation greenness (EVI), over the period 2001–2022.

The time series analysis (**Fig. A07**) reveals a coherent pattern in all subregions. Typically, positive rainfall anomalies (which automatically decrease FWI) are followed by increased EVI, indicating vegetation growth and fuel accumulation. When this is then followed by elevated FWI values (due to negative rain and humidity anomalies, extreme heat and/or strong winds), peaks in BA are frequently observed. This pattern supports the interpretation of a fire-favoring sequence: moisture enables biomass build-up, which is later dried and made flammable under high fire-weather conditions, culminating in fire activity. This cycle is particularly evident in major fire years such as 2020 and 2022, especially in the Wet Chaco, where the alignment between environmental anomalies and BA peaks is striking. In the Dry and Very Dry Chaco, the sequence is also well defined, although slightly more variable probably due to limited fuel accumulation.

The influence of large-scale climate variability, particularly the El Niño–Southern Oscillation (ENSO), is also reflected in the fire–environment dynamics. During La Niña phases (negative ENSO), we observe reduced rainfall and elevated FWI values, often coinciding with increased BA. Conversely, El Niño





453    episodes (positive ENSO) are associated with wetter conditions, lower fire-weather pressure, and

454    reduced fire activity (**Fig. A7** and **Fig. A8**).

455

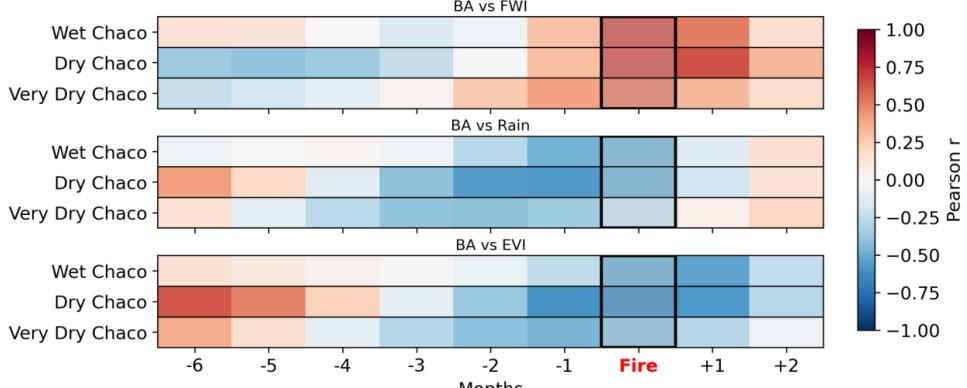

456

**Fig. 9.** Lagged correlations between monthly anomalies of FWI, rainfall, and EVI with burned area in the Chaco. Each heatmap shows the Pearson correlation coefficient between the anomaly of a given variable (FWI, rainfall, or EVI) at different time lags and the burned area anomaly, for each Chaco subregion. Negative lags indicate the variable leads burned area; positive lags indicate it follows. Correlations are computed from pixel-based, region-averaged monthly time series for 2001–2022.


**Fig. 9** shows lagged Pearson correlations between monthly anomalies of BA and FWI, rainfall, and EVI
for the three Chaco subregions. Positive correlations between BA and FWI at lags 0 to +1 months,
indicate that peak fire activity coincides with high fire-weather conditions. Rainfall and EVI display
negative correlations with BA at short negative lags (−1 to −3 months), consistent with dry, senescent
vegetation promoting flammability. At longer negative lags (−5 to −6 months), especially in the Dry and
Very Dry Chaco, both variables correlate positively with BA, suggesting that wetter, greener periods
months earlier promote fuel build-up. In the Wet Chaco, lag correlations are weaker and less structured,
likely due to consistently moist conditions that buffer fire–environment coupling.











**3.5 Fire-weather types**

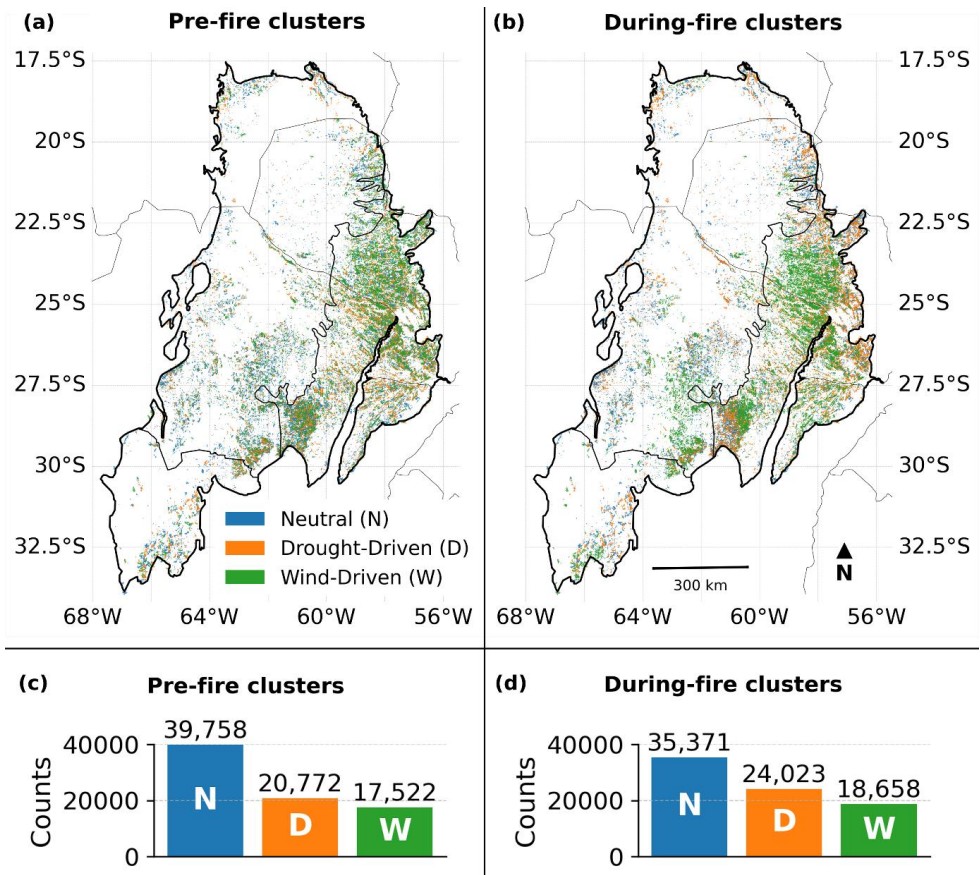


**Fig. 10.** Spatial distribution and frequency of pre- and during-fire meteorological clusters across the Gran Chaco
(2001–2022). Panels (a) and (b) show the geographic location of fire patches classified into three Fire–Weather Types
(FWTs)—Neutral (blue), Drought-Driven (orange), and Wind-Driven (green)—for the pre-fire and during-fire periods,
respectively, overlaid on Chaco sub-region boundaries Some patches overlap through the years and may partially or
totally cover each other. Panels (c) and (d) display the total number of patches assigned to each FWT for pre-fire and
       during-fire clustering methods, respectively.


**Fig. 10** shows the spatial distribution and frequency of three Fire–Weather Types (FWTs)—Neutral,
Drought-Driven, and Wind-Driven—for the pre-fire and during-fire periods. Using k-means clustering
with $k = 3$, each FP was assigned an FWT twice: first based on conditions in the 0–3 days before ignition
(*Pre-Fire*) and then based on mean conditions during the active burning period (*During-Fire*).
Neutral FWTs dominate both clusterings, but their share decreases from 50.9 % to 45.3 % overall, while
Drought-Driven rises from 26.6 % to 30.8 % and Wind-Driven from 22.4 % to 23.9 % (**Fig. 10c–d and**
**Fig. A9**). In the Wet Chaco, Neutral drops from **49 %** to **42 %** with a marked increase in Drought-
Driven; in the Dry Chaco, both non-neutral types grow moderately; in the Very Dry Chaco, Wind-



Driven increases sharply (**15 % → 26 %**), especially in the south where complex topography may
strongly influence fire–atmosphere dynamics (see **Section 2.1**).

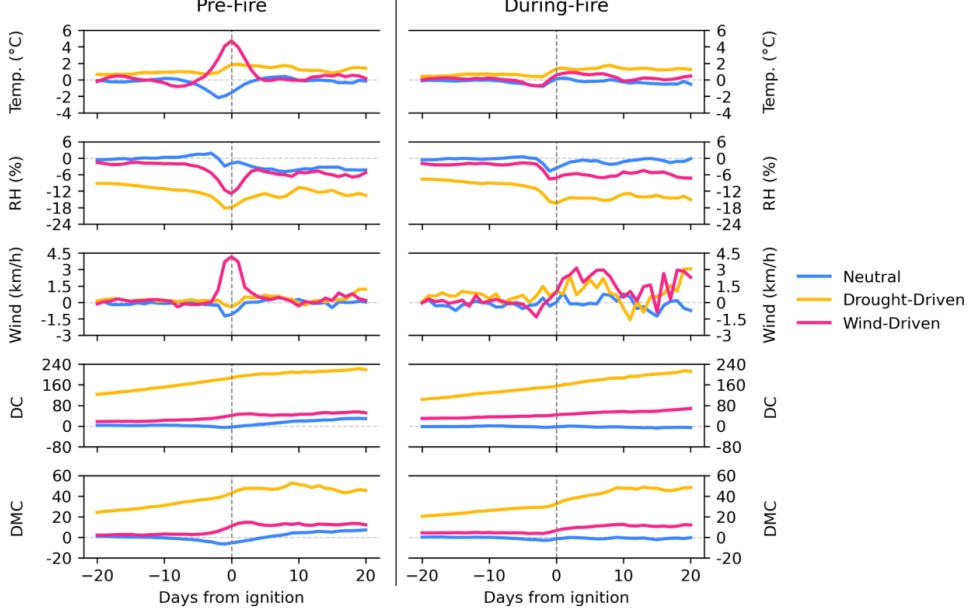


**Fig. 11.** Mean daily anomalies of temperature (Temp.), relative humidity (RH), 10-meter wind speed, Drought Code (DC), and Duff Moisture
Code (DMC) from 20 days before to 20 days after fire ignition, averaged over fire polygons assigned to the Neutral, Drought-Driven, and
Wind-Driven clusters for Pre-Fire (left) and During-Fire (right) clustering approaches.

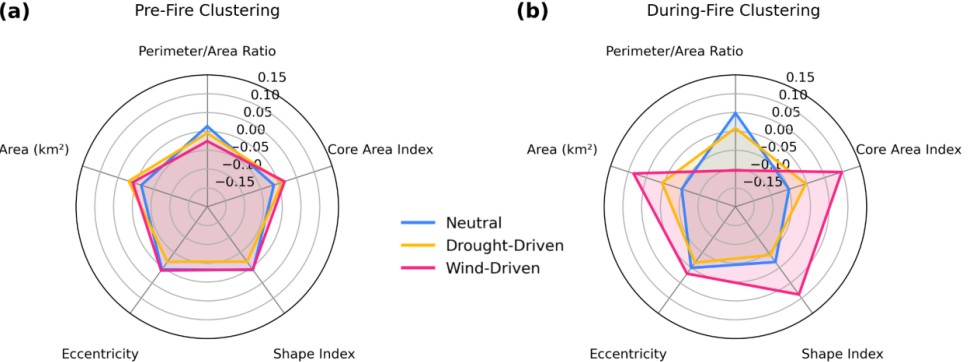


**Fig. 12.** Clusters mean morphology profiles for (a) Pre-Fire and (b) During-Fire clusterings. Each axis represents a standardised morphology
variable (z-score), and each colored polygon shows the mean profile for one cluster. The radial extent indicates the relative value of each
variable within the dataset.

**Fig. 11** shows mean daily anomalies from 20 days before to 20 days after ignition for each FWT. Wind-
Driven fires present a sharp rise in wind speed and temperature in the days around ignition, coupled with

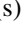


a drop in RH, creating highly flammable conditions. Drought-Driven fires exhibit a long build-up of
dryness before ignition, with persistently high DC and DMC values and low RH, indicating extended
fuel curing. Neutral fires occur under conditions close to climatology, with only small fluctuations in all
variables.
Morphology across *Pre–Fire* FWTs is broadly similar (**Fig. 12**, **A10–A11**), with comparable FP area,
shape index, core-area index, eccentricity, and perimeter-to-area ratio.
In contrast, *During-Fire* FWTs display clear differences: Wind-Driven fires tend to be larger, more
elongated, and more cohesive (higher core-area index, lower perimeter-to-area ratio) than Drought-
Driven fires, consistent with directional spread under sustained winds.
Overall, *Pre-Fire* FWTs capture the atmospheric context leading to ignition, whereas *During-Fire*
FWTs better reflect the conditions that shape the eventual size and geometry of the burned area. Other
factors such as fuel continuity, topography, and human interventions likely modulate these outcomes.

**3.6 Fire size drivers**
To identify drivers of fire size and shape beyond meteorological conditions, we trained Random Forest
(RF) models using 17 landscape and environmental predictors for all FPs between 1 km² and 100 km²
(**see Section 2.3.5**).

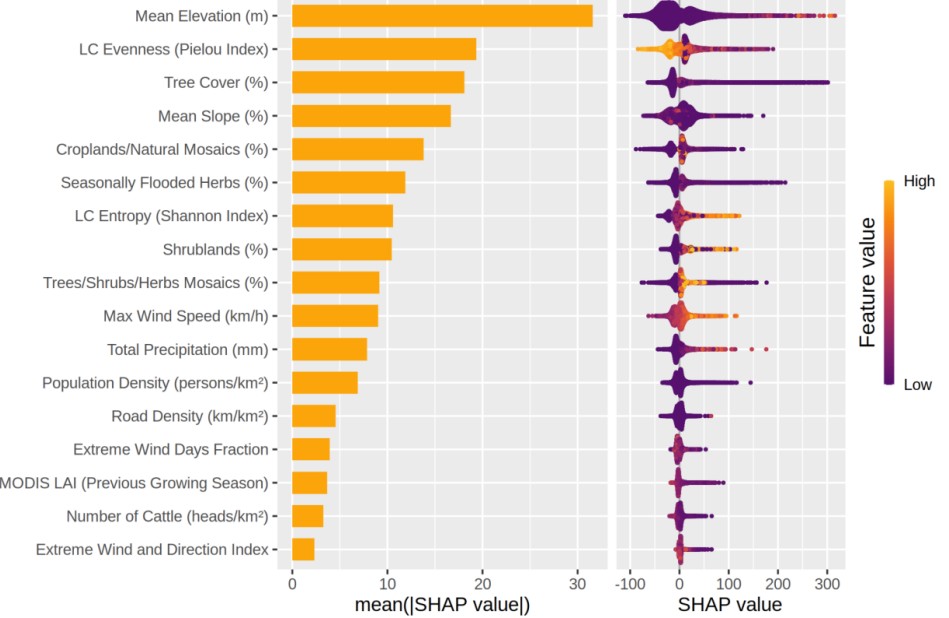


**Fig. 13.** SHAP summary plot for the Random Forest model predicting fire polygon size (n_cell) using all fire patches in the FRY dataset with
areas between 1 km² and 100 km², with 17 explanatory features extracted for each polygon. The left panel shows the mean absolute SHAP



value for each feature, ranking them by overall importance. The right panel displays the distribution of SHAP values for each feature across
all observations, with color indicating the feature value (purple = low, yellow = high).


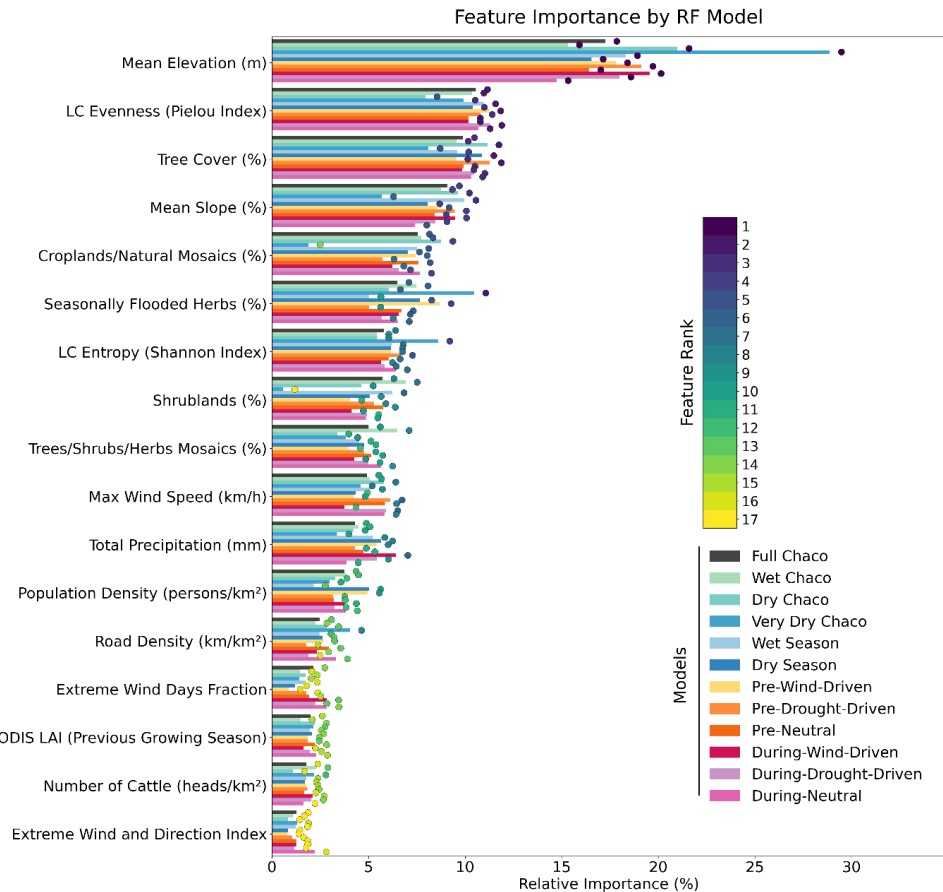


**Fig. 14.** SHAP feature importance ranks across all trained Random Forest models used to predict fire polygon size (n_cell) based on 17
explanatory variables. Colored dots at the end of bars shows the rank of a variable's importance (1 = most important, 17 = least important) for
a given model.

In the global RF model (**Fig. 13**), static topographic and vegetation structure variables dominated: mean
elevation had the highest mean SHAP value (31.3), followed by land-cover (LC) evenness (21.0), tree
cover (19.3) and mean slope (15.2). These four variables consistently ranked in the top positions across
all twelve cluster-specific and global models (**Fig. 14**). Land-cover composition metrics such as
cropland or flooded herbaceous cover showed moderate contributions, while meteorological and social
variables (e.g. maximum wind speed, precipitation, population or cattle density) were surprisingly of
lower importance.



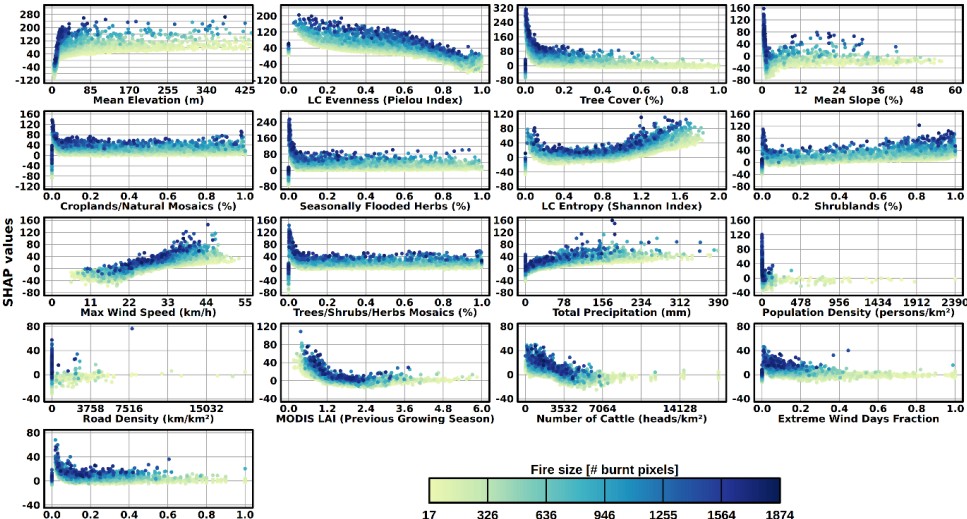

**Fig. 15.** SHAP dependence plots for all 17 explanatory variables used to predict fire polygon size (n_cell) with the Random Forest model trained on all fire patches between 1 km² and 100 km². Each panel shows the SHAP value (y-axis) across the range of a given feature (x-axis), illustrating the marginal effect of that feature on the model's output. Dots are colored by fire size (number of burned pixels), with darker tones indicating larger fires.

SHAP dependence plots (**Fig. 15**) revealed clear non-linear responses. Elevation had a steep positive effect up to ~70 m, plateauing thereafter, suggesting that slightly elevated terrain favors larger fires, while low-lying areas may be constrained by water bodies or vegetation type. Slope effects were similar: flat to gently undulating terrain (≤10 %) supported larger fires, while steeper slopes curtailed spread. Lower LC evenness (i.e. more homogeneous fuels) and sparse tree cover were associated with larger predicted sizes, reflecting the role of fuel continuity and open vegetation in promoting spread; conversely, heterogeneous landscapes and dense tree cover dampened fire growth.

Most other predictors showed weak or flat SHAP responses. Only maximum wind speed displayed a consistent positive association with fire size among the dynamic variables, indicating a secondary but detectable influence compared with dominant topographic and structural gradients.



**4 DISCUSSION**
Building on event-level fire polygons (FPs), we examine how meteorology, landscape structure, and
human pressures shape fire size and morphology across the Wet, Dry, and Very Dry Chaco.
**4.1 Fire regime and extreme events**
FP data reveal a strongly skewed size distribution: many small fires (<5 km²) and a few very large events
that dominate burned area (BA), consistent with global patterns (Archibald et al., 2009; García et al.,
2022b; Haas et al., 2022; Hantson et al., 2015, 2017). Megafires (>100 km²) are most frequent in the
Wet Chaco, where continuous herbaceous fuels in savannas and seasonally flooded vegetation support
spread. Gigafires (>1000 km²), although rare, occur almost exclusively in the drier subregions, often in
remote areas with limited suppression access, higher shrub biomass, and lower humidity. In extreme
years such as 2019–2022, a handful of these events contributed a substantial share of total BA in their
respective regions.
These size patterns indicate that both fuel configuration and atmospheric conditions influence the
potential for very large fires. We therefore examined how short-term fire weather relates to BA across
subregions. Fire weather–BA coupling shows marked spatial variability: in the Wet Chaco, high FWI is
consistently associated with large BA, confirming moisture limitation and strong sensitivity to
atmospheric conditions, in line with earlier BA-based analyses (San Martín et al., 2023). In the Dry and
Very Dry Chaco, correlations are weaker and more heterogeneous, indicating partial decoupling
between short-term fire weather and final size, mediated by fuel continuity and antecedent conditions.
Lagged relationships clarify this contrast: in drier areas, positive rainfall and vegetation productivity 4–
6 months before fire are followed by higher BA once fuels cure, supporting the fire–productivity
hypothesis (Pausas and Bradstock, 2007). In wetter areas, where fuels are rarely limiting, short dry spells
immediately prior to fire are more predictive of activity, consistent with a moisture-limited regime
within varying-constraint frameworks across resource gradients (Krawchuk and Moritz, 2011).

**4.2 Fire-weather types across the Chaco region**
To assess how daily fire weather influences fire size, we built on the framework of Hernandez et al.
(2015) and Ruffault et al. (2016, 2020), who classified Mediterranean wildfires into Fire-Weather Types
(FWTs) based on pre-fire meteorological anomalies (heat, drought, wind) and found that Hot-Drought
and Wind-Driven types were strongly linked to large events. Applying a similar pre-fire clustering in
the Gran Chaco (Neutral, Drought-Driven, Wind-Driven) captured ignition contexts but explained little
variation in final size or shape.
In contrast, clustering based on during-fire variables (maximum wind speed, total precipitation, drought
indices, and the Extreme Wind Directionality Index developed in this study) clearly separated groups



with significant differences in size and morphology. Dry, windy days during the fire, favored rapid and
large expansion.
Our findings contrast with Ruffault et al. (2016, 2020) and Belhadj-Kheder et al. (2020), who found pre-
fire or near-ignition anomalies predictive in Mediterranean and North African settings, respectively,
with the latter highlighting anomaly duration in low-suppression contexts. This stronger size–weather
link for during-fire meteorology likely reflects Chaco-specific traits such as flat terrain, continuous fuels,
and permissive fire conditions (Bucher, 1982; Vidal-Riveros et al., 2023), which make wind and
humidity more decisive than pre-fire anomalies. In the Mediterranean, fragmented fuels, complex
topography, and strong suppression (Ruffault and Mouillot, 2015, 2017), translate into ignition-day
extremes mattering more. Similar modulation by suppression capacity occurs in western U.S. forests
(Higuera et al., 2015).
Our clustering extends fire-weather typologies to a tropical dry forest context and complements recent
Gran Chaco regime classifications (Vidal-Riveros et al., 2024; Naval-Fernández et al., 2025) that
omitted meteorological variables, highlighting the key role of fire-active weather in shaping fire
morphology.
Separately, our results also showed that La Niña phases, characterised by precipitation deficits in the
Gran Chaco, coincided with elevated FWI, higher BA, and a greater likelihood of large fire events. This
pattern was particularly evident during the extreme fire seasons of 2019–2022, illustrating how
interannual climate variability modulates fire size potential at regional scales.

## 4.3 Landscape pattern influence on fire types

Beyond meteorological effects, anthropogenic and structural landscape factors strongly modulated fire
size. Random Forest (RF) models consistently identified elevation as the most important predictor across
all subregions and seasons, followed by land-cover evenness, tree cover, and slope (Fig. 14). While
elevation is not a direct control on combustion, it reflects broad ecological gradients in vegetation
composition, fuel moisture regimes, and land-use history that shape the conditions under which fires
develop. In the Chaco, these gradients often translate into water presence and seasonal flooding in
lowlands, which can limit spread, and stronger, more persistent winds in higher terrain, which can
enhance it.
Vegetation composition exerted a strong influence on size outcomes. Areas dominated by herbaceous
or shrub cover, often linked to past or ongoing land-use change, were more prone to large fires, whereas
higher tree cover was associated with smaller fires. This pattern aligns with global evidence that
increasing tree cover generally reduces burned area (Bistinas et al., 2014; Haas et al., 2022), although
exceptions occur where certain forest types, such as introduced pine plantations, have higher
flammability than native broadleaf evergreen forests (Barros and Pereira, 2014; Paritsis et al., 2018;



Vidal-Riveros et al., 2023). Differences in live fuel moisture between growth forms (Yebra et al., 2019)
further explain the greater spread potential in shrub- and grass-dominated systems.
Landscape heterogeneity, expressed as lower land-cover evenness (i.e., more homogeneous fuels), was
also linked to larger fires, reflecting the role of continuous fuel beds in enabling propagation.
Conversely, heterogeneous mosaics with high evenness disrupted spread, acting as natural firebreaks
(Povak et al., 2018). Together, these results show that while fire-active weather is an important
determinant of spread (**Section 4.2**), the physical and vegetative structure of the landscape sets the upper
limits for how large fires can become.

**4.4 Fire shape as an indicator of fire weather**
Building on the fire-weather clustering (**Section 4.2**) and landscape controls (**Section 4.3**), we examined
whether fire morphology can reveal the influence of landscape or climatic drivers of spread, taking
advantage of the detailed FP-level shape and size metrics provided by FRYv2.0 (Laurent et al., 2018;
Chen, 2025). We hypothesized that elongation and perimeter complexity would be enhanced by strong,
steady winds, whereas complex topography or fragmented fuels would produce more irregular shapes.
In the Gran Chaco, fires occurring under strong, persistent winds displayed significantly larger
perimeters and greater elongation, supporting our hypothesis and highlighting morphology as a signature
of wind-driven fire types.
To our knowledge, the hypothesis that fire elongation and perimeter complexity can serve as indicators
of prevailing wind influence on fire spread has rarely been tested directly, making this a novel
contribution of our study. Barros et al. (2012, 2013) showed that watershed orientation influenced fire
spread in California, and Mansuy et al. (2014) reported similar effects in Canadian boreal forests, but
neither explicitly linked shape to dominant wind direction. We propose that the combined analysis of
shape and size offers a valuable benchmark for process-based fire models, which often rely on simplified
ellipsoidal spread assumptions (Hantson et al., 2016), and could help train emerging machine-learning
approaches for global fire hazard prediction (Li et al., 2023; Liu et al., 2025; Zhang et al., 2023).

**4.5 Deforestation and Prescribed Burning**
Anthropogenic influences on the Gran Chaco fire regime include the advancing agricultural frontier,
characterized by rapid land-use change and deforestation (Arriaga Velasco-Aceves et al., 2021; Boletta
et al., 2006), and the widespread use of fire as a management tool. Prescribed burning typically occurs
in late winter and early spring, before the wet season (San Martín et al., 2023), and is generally limited
to periods with lower wind speed and limited drought, following decision-support guidelines for ignition
(Hsu et al., 2025). However, forecasts are uncertain, and fire-prone conditions can quickly develop after



ignition, allowing burns to escape their intended boundaries. Such escaped prescribed fires, although
often managed to limit societal impacts, remain a recurrent hazard (Black et al., 2020; Li et al., 2025).
FRYv2.0 and other global burned-area products cannot distinguish between wildfires and prescribed
burns, restricting our ability to assess their occurrence in the region. Although Hsu et al. (2025) compiled
a global prescribed fire dataset, the Gran Chaco is not covered. Many spring fires are likely prescribed
burns, but systematic monitoring is lacking. Similarly, we could not isolate deforestation fires, which in
the region tend to occur mostly within three years after forest clearing (San Martín et al., 2023). High-
resolution burned-area products combined with tree-cover data could help identify such events, as
demonstrated for Africa (Khairoun et al., 2024).
Improved detection of prescribed and deforestation fires would enable better risk assessment of escaped
burns and could promote greater societal acceptance of prescribed fire as part of integrated fire
management for hazard mitigation (Oliveras Menor et al., 2025).

**4.6 Limitations**
Direct human influences, such as ignition sources, suppression actions, and fire management practices,
could not be explicitly included in this study due to limited data availability. Their effects are likely
reflected indirectly through variables such as vegetation structure, road density, population density, and
land cover, but their absence restricts our ability to fully capture anthropogenic modulation of fire size.
The ERA5-Land reanalysis at 0.1° (~9 km) resolution, although considered high for a global
meteorological dataset, remains too coarse to fully represent local-scale wind variability, solar radiation
heterogeneity, and terrain-induced thermal gradients that can influence fire spread. Advances in
downscaling techniques for wind (Dujardin and Lehning, 2022), solar radiation (Druel et al., 2025), and
temperature (Kusch and Davy, 2022) may improve the spatial realism of these variables in future fire
regime analyses, especially in complex landscapes. However, these approaches were not applied here.
More fundamentally, the absence of dynamic coupling between fire behaviour and atmospheric
processes remains a key constraint, as fire–atmosphere feedbacks are not represented in our predictors.
The FRYv2.0 fire dataset is based on the global 250 m FireCCI51 product, which can both overestimate
and underestimate fire size. Overestimation may occur when partially burned pixels are classified as
fully burned, particularly along fire edges or within heterogeneous scars (Pettinari et al., 2021).
Underestimation arises from omission errors, which are common for small, low-intensity, or fragmented
fires that fall below the detection threshold, or in areas affected by cloud cover, dense smoke, or mixed
land cover (Lizundia-Loiola et al., 2022).
Other FireCCI51-specific limitations should also be acknowledged. BA is likely underestimated during
the early period of the dataset (2001 to mid-2002) when only Terra MODIS data were available. Ignition
dates may contain biases depending on satellite detection quality and meteorological conditions
(Lizundia-Loiola et al., 2020). Furthermore, the aggregation of pixels into FPs depends on temporal



thresholds used to group neighbouring pixels within the same event (Moreno et al., 2021; Oom et al.,

706    2016).

Future developments in fine-resolution burned-area products (e.g. 20 m), such as FireCCISFD20, have
already demonstrated substantial improvements in Africa, detecting 80–120 % more burned area
(Chuvieco et al., 2022). Delivering similar products at continental or global scale, as long requested by
the fire science community (Mouillot et al., 2014), will be critical to reduce both overestimation from
coarse-pixel classification and underestimation from omission errors, and to improve the accuracy of
fire size and distribution assessments.





## 5 CONCLUSIONS

This study advances understanding of fire regimes across the Wet, Dry, and Very Dry Chaco through a spatially explicit analysis of fire events from 2001–2022. We document strong regional contrasts in fire size, seasonality, and drivers, shaped by interactions between fuels, weather, and land use.

Fire sizes were highly skewed: over 80% of detected fires were <5 km², yet large events dominated burned area (BA). Megafires (>100 km²) occurred in all subregions, with the Wet Chaco recording the most. Gigafires (>1000 km²) were rare but concentrated in the Dry Chaco, where some single events exceeded 50% of annual BA. The Wet Chaco burned most extensively (~2× the Dry Chaco), with the highest fire frequency and ignition density, reflecting greater biomass productivity and continuous fuels. The Fire Weather Index (FWI) showed its strongest, most coherent relationship with BA in the Wet Chaco (r up to 0.7), while drier subregions displayed weaker, more heterogeneous patterns, indicating additional controls. The 2020–2022 drought produced unprecedented fire activity, though large outbreaks also occurred without extreme FWI, underscoring the role of ignition patterns and fuel availability. In the Wet Chaco, 93% of pixels had positive FWI–fire correlations, compared to ~60% in the Dry and Very Dry Chaco.

Lag analyses revealed dual mechanisms: in drier areas, wet-season biomass buildup (4–6 months prior) preceded high fire activity, while in wetter areas, short-term pre-fire dryness was more predictive. La Niña phases amplified fire potential via reduced rainfall and elevated FWI.

During-fire clustering of fire-weather types (FWTs) identified wind intensity and directionality as stronger predictors of fire morphology than other pre-fire conditions. Persistent winds produced larger, elongated, and cohesive burns, highlighting morphology as an indicator of wind-driven dynamics.

Random Forest models ranked mean elevation, land cover evenness, tree cover, and slope highest in size prediction. Larger fires occurred in flat, low-elevation areas with low tree cover; steeper slopes and higher forest cover limited spread.

In the Dry and Very Dry Chaco, part of the BA comes from one-time deforestation fires occurring after clearing, generally small to moderate in size. Extreme megafires and gigafires instead resulted from rare alignments of continuous fuels and exceptional weather, especially persistent winds and prolonged dryness, which exceeded suppression capacity. This distinction is critical for separating land-use-related burns from large climatic extremes in risk assessments.

By combining medium-resolution fire patch data, reanalysis-based weather metrics, machine learning, and landscape analysis, we identify key biophysical, climatic, and anthropogenic determinants of fire size and shape. These findings inform fire risk forecasting and management under ongoing land-use intensification and climate variability, and highlight the potential of morphology and during-fire wind metrics to benchmark and improve process-based global fire models.



749    **6 APPENDIX A**

750

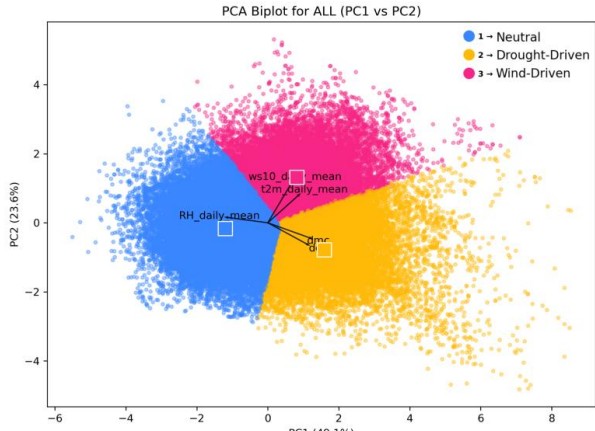

751

**Fig. A1:** Principal Component Analysis (PCA) biplot of pre-fire meteorological anomalies used for K-means clustering, showing the
distribution of fire patches across the first two principal components (PC1 and PC2), which explain 49.1% and 23.6% of the total variance,
respectively. The three clusters are color-coded and numbered as follows: Cluster 1 (blue) corresponds to Neutral conditions, Cluster 2 (orange)
to Drought-Driven conditions (with high DC and DMC anomalies), and Cluster 3 (pink) to Wind-Driven conditions (characterized by elevated
wind speed and temperature anomalies). Arrows represent the contribution of the original variables to the PCA axes. This ordination was used
to guide the semantic naming of clusters.


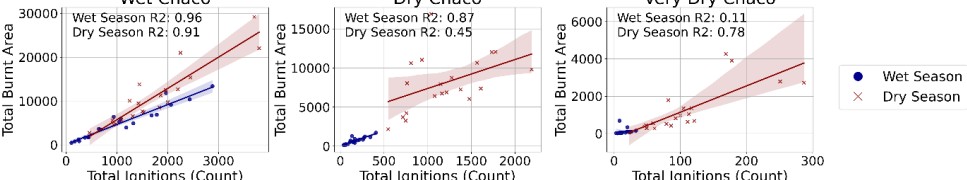


**Fig. A2:** Scatter plots and linear regressions between total annual BA and total annual ignitions between 2001 and 2022 in the Wet, Dry and
Very Dry Chaco, divided into wet season fires (blue circles) and dry season fires (red crosses).



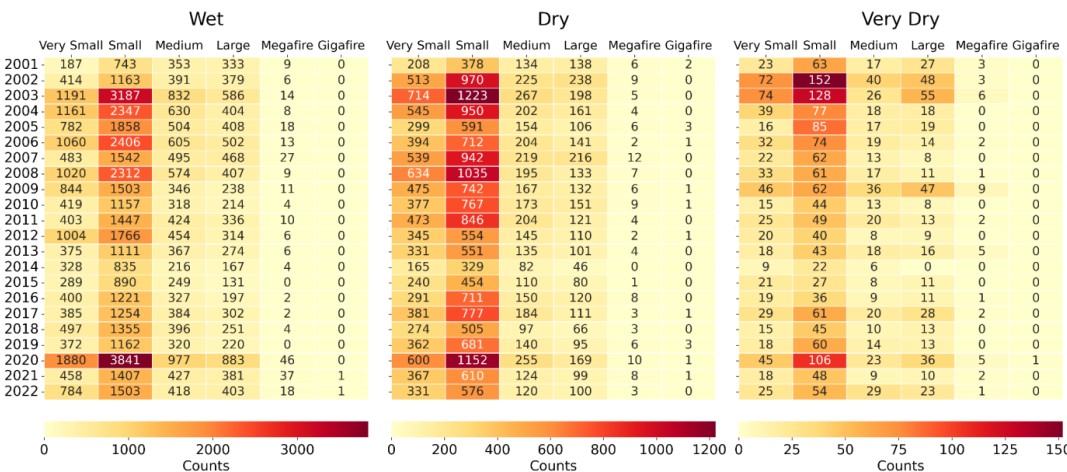

**Fig. A3:** Total counts of fire polygons separated by size category between 2001 and 2022 in the Wet, Dry, and Very Dry Chaco.

**Table A1.** Number of fires detected by FRYv2.0 between 2001 and 2022 classified by fire size. WS: wet season; DS: dry season.

| Region | Very Small (0-1 km²) | | Small (1-5 km²) | | Medium (5-10 km²) | | Large (10-100 km²) | | Megafire (100-1000 km²) | | Gigafire (> 1000 km²) | | Total |
|---|---|---|---|---|---|---|---|---|---|---|---|---|---|
| Season | WS | DS | WS | DS | WS | DS | WS | DS | WS | DS | WS | DS | |
| Wet | 8414 | 6322 | 17,018 | 18,992 | 4340 | 5667 | 3264 | 4534 | 91 | 163 | 2 | 0 | 68,807 |
| | 14,736 | | 36,010 | | 10,007 | | 7,798 | | 254 | | 2 | | |
| Dry | 3526 | 5332 | 5754 | 10,302 | 1201 | 2485 | 841 | 1991 | 24 | 94 | 0 | 15 | 31,565 |
| | 8,858 | | 16,056 | | 3,686 | | 2,832 | | 118 | | 15 | | |
| Very Dry | 334 | 300 | 708 | 691 | 187 | 203 | 200 | 238 | 13 | 29 | 0 | 1 | 2,904 |
| | 634 | | 1,399 | | 390 | | 438 | | 42 | | 1 | | |
| Total | 24,228 | | 53,465 | | 14,083 | | 11,068 | | 414 | | 18 | | 103,276 |



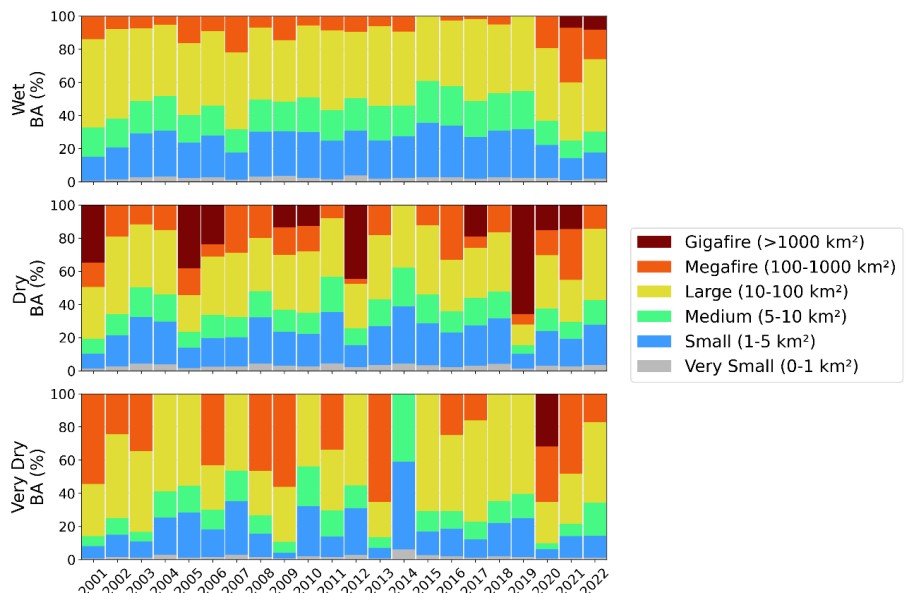


**Fig. A4:** Annual percentage distribution of burned areas across different size categories between 2001 and 2022 in the Wet, Dry, and Very Dry
Chaco.

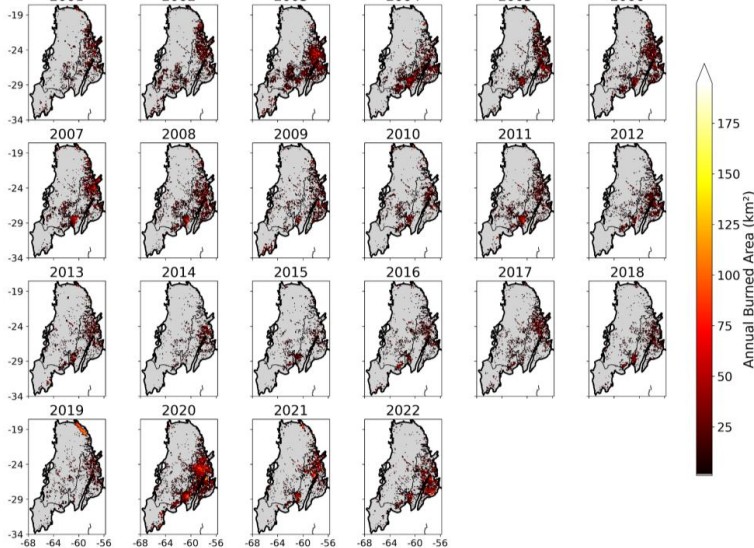


**Fig. A5:** Annual burned area maps of the Chaco region between 2001 and 2022. Burned areas extracted from FRYv2.0.





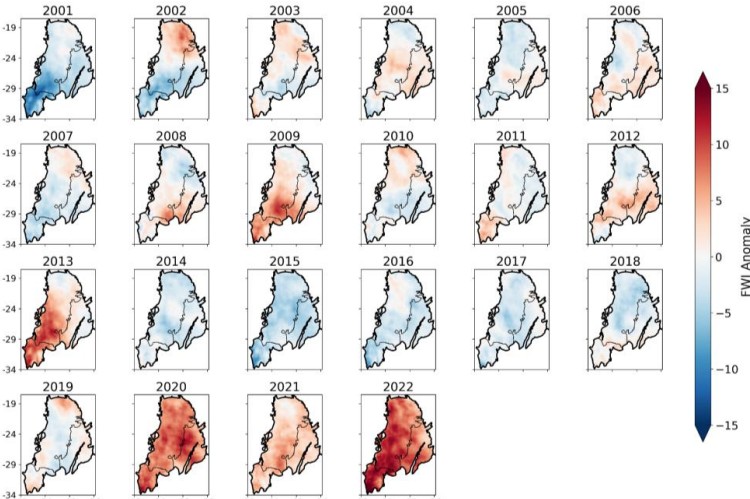

**Fig. A6:** Annual mean Fire Weather Index (FWI) anomalies with respect to the period 2001–2020, averaged for the Chaco region for each year between 2001 and 2022. FWI built from ERA5-Land.

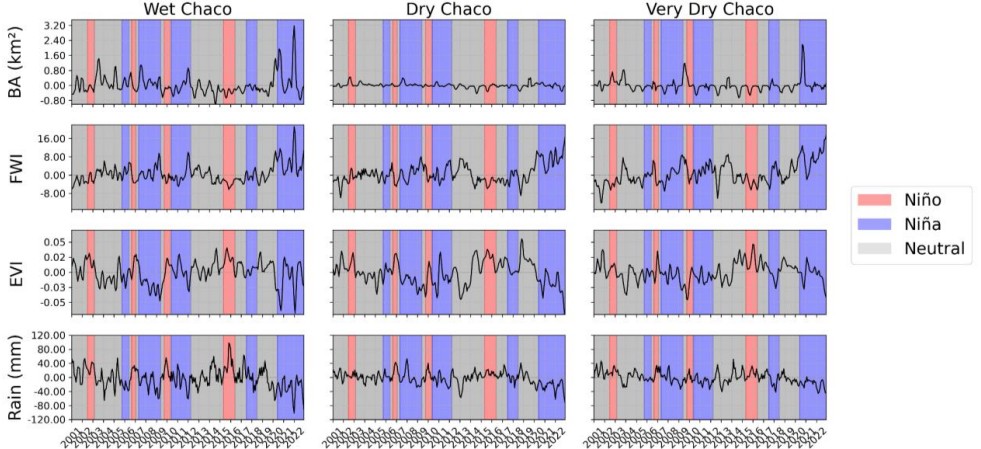

**Fig. A7:** Monthly anomalies of rainfall, vegetation (EVI), fuel dryness (FWI), and burned area in the Chaco subregions. Panels show 3-month running means of region-averaged anomalies for each variable, calculated from gridded (pixel-based) data and averaged over the Wet, Dry, and Very Dry Chaco subregions. Shaded backgrounds in the burned area panel indicate ENSO phases (red for El Niño, blue for La Niña), calculated with the Multivariate ENSO index (MEI).




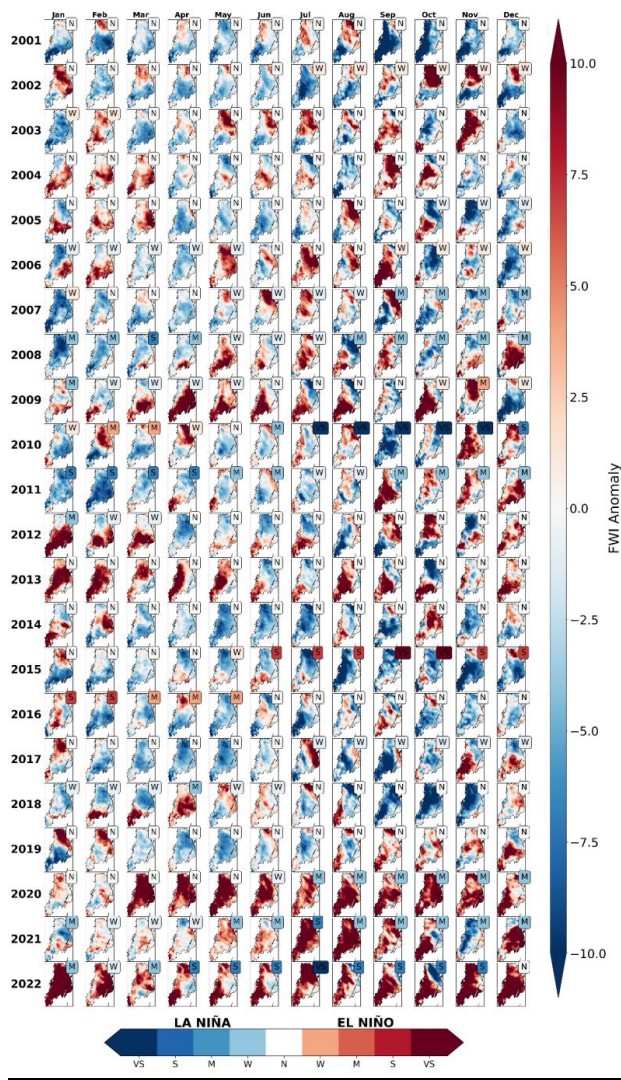


**Fig. A8:** The maps display the monthly anomalies (with 2001–2021 as the baseline) for the Chaco region for each year within the period. Additionally, each map counts with the Multivariate ENSO Index (MEI) showing the presence of an El Niño (EN; red) or La Niña (LN; blue) when during five consecutive three-month periods, MEI values are above +0.5 or below -0.5, respectively. Otherwise, the months are in a neutral (N) phase. The Niño/Niña events are classified by intensity based on the absolute MEI values. W: Weak (≥ 0.5); M: Moderate (≥ 1); S: Strong (≥ 1.5); VS: Very Strong (≥ 2).





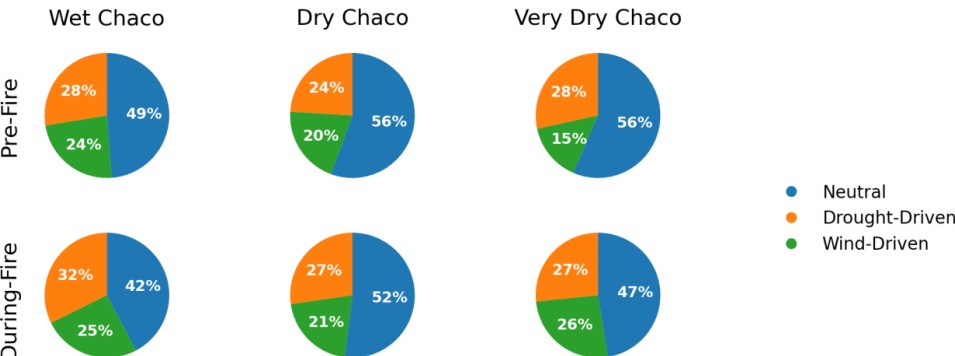

**Fig. A9:** Regional distribution of fire-weather types (FWTs) across the three Chaco subregions based on the Pre-Fire clustering (top row) and the During-Fire clustering (bottom row). Pie charts represent the proportion of fire patches assigned to each cluster—Drought-Driven (orange), Wind-Driven (green), and Neutral (blue)—based on pre-fire (0–3 days before ignition) and during-fire meteorological conditions.

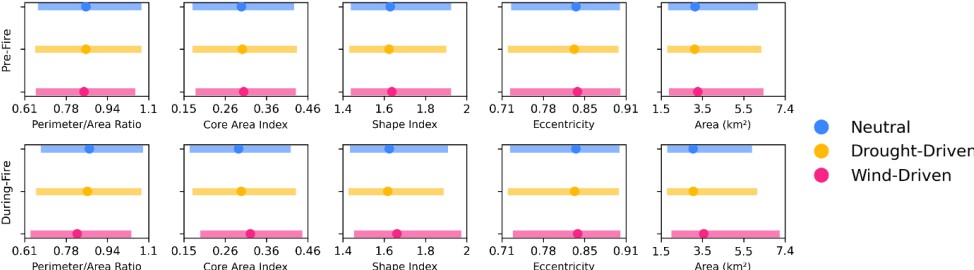

**Fig. A10:** Distribution of morphology variables by cluster (quartile–dot plots). For each morphology variable, the interquartile range (IQR; thick horizontal bar) and median (dot) are shown for each cluster, separately for Pre-Fire and During-Fire clusterings (first and second rows, respectively). This visualizes the spread and central tendency of each variable within clusters, highlighting differences in fire patch morphology between cluster types and fire periods.

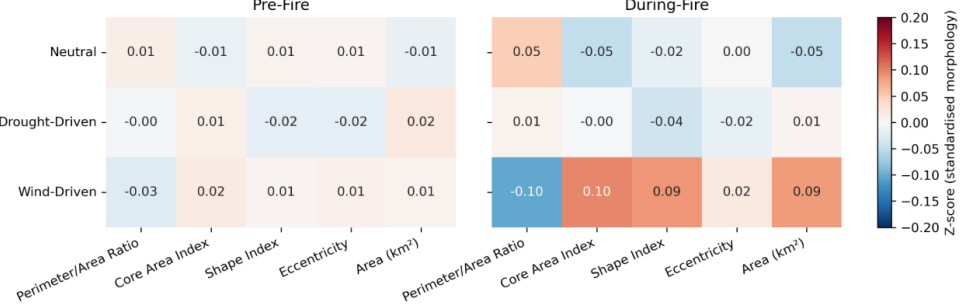

**Fig. A11:** Each heatmap shows the mean z-score (standardised value) of key fire patch morphology variables for each cluster, separately for Pre-Fire (left) and During-Fire (right) cluster assignments. Rows correspond to clusters (Neutral, Drought-Driven, Wind-Driven), and columns to morphology variables. The color scale indicates the relative position of each cluster's mean within the overall distribution, highlighting differences in fire patch shape and size between clusters and fire periods.



## 7 AUTHOR CONTRIBUTION

RSM collected and processed the data, analyzed the results, and drafted the manuscript. CO and AS conceived the idea and led the project. PVA contributed to data analysis, specifically by performing Random Forest modeling. All co-authors discussed the results, provided critical feedback, and reviewed the manuscript.

## 8 COMPETING INTERESTS

The authors declare that they have no conflict of interest.

## 9 ACKNOWLEDGEMENTS

The authors thank all the researchers and institutions involved in providing open-access datasets, including ESA CCI, ERA5-Land, and the Copernicus Climate Data Store (CDS). We acknowledge the computational infrastructure and support provided by the Laboratoire des Sciences du Climat et de l'Environnement (LSCE/IPSL). We also express our gratitude to Dr. Sandra Bravo for her important collaboration and contributions to our understanding of the fire regime in the region, as well as colleagues from CONICET for their valuable insights into Chaco ecology. The authors also acknowledge the use of AI-based tools to assist with text editing, code debugging, and figure scripting throughout the preparation of the manuscript.

## 9 FINANCIAL SUPPORT

This research was partially funded by the European Space Agency through the Climate Change Initiative programme, under contract numbers ESA/No. 4000126564 (Land_Cover_cci) and ESA ESRIN/No. 4000125259/18/I-NB. A. Sörensson acknowledges support from the Agencia Nacional de Promoción Científica y Tecnológica (ANPCyT, Argentina) via project PICT 2018-02511, and from the Consejo Nacional de Investigaciones Científicas y Técnicas (CONICET, Argentina) through grant PIP 11220200102141CO. R. San Martin received doctoral funding from the Environmental Science Doctoral School of Île-de-France (DS 129).





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

Contributions to Wildland Fire Science and Management, Curr. For. Rep., 6, 81–96,
https://doi.org/10.1007/s40725-020-00116-5, 2020.



Chuvieco, E., Roteta, E., Sali, M., Stroppiana, D., Boettcher, M., Kirches, G., Storm, T., Khairoun, A.,
Pettinari, M. L., Franquesa, M., and Albergel, C.: Building a small fire database for Sub-Saharan Africa
from    Sentinel-2    high-resolution    images,    Sci.    Total    Environ.,    845,    157139,
https://doi.org/10.1016/j.scitotenv.2022.157139, 2022.
D'Antonio, C. M. and Vitousek, P. M.: Biological Invasions by Exotic Grasses, the Grass/Fire Cycle,
and    Global    Change,    Annu.    Rev.    Ecol.    Evol.    Syst.,    23,    63–87,
https://doi.org/10.1146/annurev.es.23.110192.000431, 1992.
De Marzo, T., Pflugmacher, D., Baumann, M., Lambin, E. F., Gasparri, I., and Kuemmerle, T.:
Characterizing forest disturbances across the Argentine Dry Chaco based on Landsat time series, Int. J.
Appl. Earth Obs. Geoinformation, 98, 102310, https://doi.org/10.1016/j.jag.2021.102310, 2021.
De Marzo, T., Pratzer, M., Baumann, M., Gasparri, N. I., Pötzschner, F., and Kuemmerle, T.: Linking
disturbance history to current forest structure to assess the impact of disturbances in tropical dry forests,
For. Ecol. Manag., 539, 120989, https://doi.org/10.1016/j.foreco.2023.120989, 2023.
Doblas-Reyes, F. J., Sorensson, A. A., Almazroui, M., Dosio, A., Gutowski, W. J., Haarsma, R., Hamdi,
R., Hewitson, B., Kwon, W.-T., Lamptey, B. L., Maraun, D., Stephenson, T. S., Takayabu, I., Terray,
L., Turner, A., and Zuo, Z.: Linking global to regional climate change, edited by: Masson-Delmotte, V.,
Zhai, P., Pirani, A., Connors, S. L., Pean, C., Berger, S., Caud, N., Chen, Y., Goldfarb, L., Gomis, M.
I., Huang, M., Leitzell, K., Lonnoy, E., Matthews, J. B. R., Maycock, T. K., Waterfield, T., Yelekci, O.,
Yu, R., and Zhou, B., Cambridge University Press, 2021.
Druel, A., Ruffault, J., Davi, H., Chanzy, A., Marloie, O., De Cáceres, M., Olioso, A., Mouillot, F.,
François, C., Soudani, K., and Martin-StPaul, N. K.: Enhancing environmental models with a new
downscaling method for global radiation in complex terrain, Biogeosciences, 22, 1–18,
https://doi.org/10.5194/bg-22-1-2025, 2025.
Dujardin, J. and Lehning, M.: Wind-Topo: Downscaling near-surface wind fields to high-resolution
topography in highly complex terrain with deep learning, Q. J. R. Meteorol. Soc., 148, 1368–1388,
https://doi.org/10.1002/qj.4265, 2022.
SRTM | Earthdata: https://www.earthdata.nasa.gov/sensors/srtm, last access: 4 September 2024.
García, M., Pettinari, M. L., Chuvieco, E., Salas, J., Mouillot, F., Chen, W., and Aguado, I.:
Characterizing Global Fire Regimes from Satellite-Derived Products, Forests, 13, 699,
https://doi.org/10.3390/f13050699, 2022a.
García, M., Pettinari, M. L., Chuvieco, E., Salas, J., Mouillot, F., Chen, W., and Aguado, I.:
Characterizing Global Fire Regimes from Satellite-Derived Products, Forests, 13, 699,
https://doi.org/10.3390/f13050699, 2022b.
Gasparri, N. I., Grau, H. R., and Manghi, E.: Carbon Pools and Emissions from Deforestation in Extra-
Tropical Forests of Northern Argentina Between 1900 and 2005, Ecosystems, 11, 1247–1261,
https://doi.org/10.1007/s10021-008-9190-8, 2008.
Ginzburg, R., Adámoli, J., Herrera, P., and Torrella, S.: Los Humedales del Chaco: clasificación,
inventario y mapeo a escala regional, Miscelánea, 14, 121–138, 2005.
Haas, O., Prentice, I. C., and Harrison, S. P.: Global environmental controls on wildfire burnt area, size,
and intensity, Environ. Res. Lett., 17, 065004, https://doi.org/10.1088/1748-9326/ac6a69, 2022.
Hantson, S., Pueyo, S., and Chuvieco, E.: Global fire size distribution is driven by human impact and
climate, Glob. Ecol. Biogeogr., 24, 77–86, https://doi.org/10.1111/geb.12246, 2015.



Hantson, S., Arneth, A., Harrison, S. P., Kelley, D. I., Prentice, I. C., Rabin, S. S., Archibald, S.,
Mouillot, F., Arnold, S. R., Artaxo, P., Bachelet, D., Ciais, P., Forrest, M., Friedlingstein, P., Hickler,
T., Kaplan, J. O., Kloster, S., Knorr, W., Lasslop, G., Li, F., Mangeon, S., Melton, J. R., Meyn, A., Sitch,
S., Spessa, A., van der Werf, G. R., Voulgarakis, A., and Yue, C.: The status and challenge of global
fire modelling, Biogeosciences, 13, 3359–3375, https://doi.org/10.5194/bg-13-3359-2016, 2016.
Hantson, S., Scheffer, M., Pueyo, S., Xu, C., Lasslop, G., Nes, E. H., and Mendelsohn, J.: Rare, Intense,
Big fires dominate the global tropics under drier conditions, Sci. Rep., 7, 1–5, 2017.
Hengl, T., Mendes De Jesus, J., Heuvelink, G. B. M., Ruiperez Gonzalez, M., Kilibarda, M., Blagotić,
A., Shangguan, W., Wright, M. N., Geng, X., Bauer-Marschallinger, B., Guevara, M. A., Vargas, R.,
MacMillan, R. A., Batjes, N. H., Leenaars, J. G. B., Ribeiro, E., Wheeler, I., Mantel, S., and Kempen,
B.: SoilGrids250m: Global gridded soil information based on machine learning, PLOS ONE, 12,
e0169748, https://doi.org/10.1371/journal.pone.0169748, 2017.
Hernandez, C., Drobinski, P., and Turquety, S.: How much does weather control fire size and intensity
in the Mediterranean region?, Ann. Geophys., 33, 931–939, https://doi.org/10.5194/angeo-33-931-2015,
979 2015.

Higuera, P. E., Abatzoglou, J. T., Littell, J. S., and Morgan, P.: The Changing Strength and Nature of
Fire-Climate Relationships in the Northern Rocky Mountains, U.S.A., 1902-2008, PLOS ONE, 10,
e0127563, https://doi.org/10.1371/journal.pone.0127563, 2015.
Hsu, A., Jones, M. W., Thurgood, J. R., Smith, A. J. P., Carmenta, R., Abatzoglou, J. T., Anderson, L.
O., Clarke, H., Doerr, S. H., Fernandes, P. M., Kolden, C. A., Santín, C., Strydom, T., Le Quéré, C.,
Ascoli, D., Castellnou, M., Goldammer, J. G., Guiomar, N. R. G. N., Kukavskaya, E. A., Rigolot, E.,
Tanpipat, V., Varner, M., Yamashita, Y., Baard, J., Barreto, R., Becerra, J., Brunn, E., Bergius, N.,
Carlsson, J., Cheney, C., Druce, D., Elliot, A., Evans, J., De Moraes Falleiro, R., Prat-Guitart, N., Hiers,
J. K., Kaiser, J. W., Macher, L., Morris, D., Park, J., Robles, C., Román-Cuesta, R. M., Rücker, G.,
Senra, F., Steil, L., Valverde, J. A. L., and Zerr, E.: A global assemblage of regional prescribed burn
records — GlobalRx, Sci. Data, 12, 1083, https://doi.org/10.1038/s41597-025-04941-w, 2025.
Jolly, W. M., Cochrane, M. A., Freeborn, P. H., Holden, Z. A., Brown, T. J., Williamson, G. J., and
Bowman, D. M. J. S.: Climate-induced variations in global wildfire danger from 1979 to 2013, Nat.
Commun., 6, 7537, https://doi.org/10.1038/ncomms8537, 2015.
Jones, Abatzoglou, J. T., Veraverbeke, S., Andela, N., Lasslop, G., and Forkel, M.: Global and regional
trends and drivers of fire under climate change, Rev. Geophys., 60, 2020 000726,
https://doi.org/10.1029/2020RG000726, 2022.
Kelley, D. I., Bistinas, I., Whitley, R., Burton, C., Marthews, T. R., and Dong, N.: How contemporary
bioclimatic and human controls change global fire regimes, Nat. Clim. Change, 9, 690–696,
https://doi.org/10.1038/s41558-019-0540-7, 2019.
Khairoun, A., Mouillot, F., Chen, W., Ciais, P., and Chuvieco, E.: Coarse-resolution burned area datasets
severely underestimate fire-related forest loss, Sci. Total Environ., 920, 170599,
https://doi.org/10.1016/j.scitotenv.2024.170599, 2024.
Krawchuk, M. A. and Moritz, M. A.: Constraints on global fire activity vary across a resource gradient,
Ecology, 92, 121–132, https://doi.org/10.1890/09-1843.1, 2011.
Krawchuk, M. A., Cumming, S. G., and Flannigan, M. D.: Predicted changes in fire weather suggest
increases in lightning fire initiation and future area burned in the mixedwood boreal forest, Clim.
Change, 92, 83–97, https://doi.org/10.1007/s10584-008-9460-7, 2009.





Kuemmerle, T., Altrichter, M., Baldi, G., Cabido, M., Camino, M., Cuellar, E., and Zak, M.: Forest
conservation: remember gran chaco, Science, 355, 465–465, 2017.
Kunst, C., Bravo, S., Monti, E., Cornacchione, M., and Godoy, J.: El fuego y el manejo de pasturas
naturales y cultivadas de la región chaqueña, Fuego En Los Ecosistemas Argent. Ediciones INTA, 21,
239–247, 2003.
Kusch, E. and Davy, R.: KrigR – A tool for downloading and statistically downscaling climate reanalysis
data, Environ. Res. Lett., 17, https://doi.org/10.1088/1748-9326/ac48b3, 2022.
Laurent, P., Mouillot, F., Yue, C., Ciais, P., Moreno, M. V., and Nogueira, J. M. P.: FRY, a global
database of fire patch functional traits derived from space-borne burned area products, Sci. Data, 5,
180132, https://doi.org/10.1038/sdata.2018.132, 2018.
Lehner, B., Verdin, K., and Jarvis, A.: New Global Hydrography Derived From Spaceborne Elevation
Data, Eos Trans. Am. Geophys. Union, 89, 93–94, https://doi.org/10.1029/2008eo100001, 2008.
Levers, C., Piquer-Rodríguez, M., Gollnow, F., Baumann, M., Camino, M., Gasparri, N. I., Gavier-
Pizarro, G. I., le Polain de Waroux, Y., Müller, D., Nori, J., Pötzschner, F., Romero-Muñoz, A., and
Kuemmerle, T.: What is still at stake in the Gran Chaco? Social-ecological impacts of alternative land-
system futures in a global deforestation hotspot, Environ. Res. Lett., 19, 064003,
https://doi.org/10.1088/1748-9326/ad44b6, 2024.
Li, F., Zhu, Q., Riley, W. J., Zhao, L., Xu, L., Yuan, K., Chen, M., Wu, H., Gui, Z., Gong, J., and
Randerson, J. T.: AttentionFire_v1.0: interpretable machine learning fire model for burned-area
predictions over tropics, Geosci. Model Dev., 16, 869–884, https://doi.org/10.5194/gmd-16-869-2023,
1028  2023.

Li, S., Baijnath-Rodino, J. A., York, R. A., Quinn-Davidson, L. N., and Banerjee, T.: Temporal and
spatial pattern analysis of escaped prescribed fires in California from 1991 to 2020, Fire Ecol., 21, 3,
https://doi.org/10.1186/s42408-024-00342-3, 2025.
Linley, G. D., Jolly, C. J., Doherty, T. S., Geary, W. L., Armenteras, D., Belcher, C. M., Bliege Bird,
R., Duane, A., Fletcher, M., Giorgis, M. A., Haslem, A., Jones, G. M., Kelly, L. T., Lee, C. K. F., Nolan,
R. H., Parr, C. L., Pausas, J. G., Price, J. N., Regos, A., Ritchie, E. G., Ruffault, J., Williamson, G. J.,
Wu, Q., and Nimmo, D. G.: What do you mean, 'megafire'?, Glob. Ecol. Biogeogr., 31, 1906–1922,
https://doi.org/10.1111/geb.13499, 2022.
Liu, Y., Huang, H., Wang, S.-C., Zhang, T., Xu, D., and Chen, Y.: ELM2.1-XGBfire1.0: improving
wildfire prediction by integrating a machine learning fire model in a land surface model, Geosci. Model
Dev., 18, 4103–4117, https://doi.org/10.5194/gmd-18-4103-2025, 2025.
Lizundia-Loiola, J., Otón, G., Ramo, R., and Chuvieco, E.: A spatio-temporal active-fire clustering
approach for global burned area mapping at 250 m from MODIS data, Remote Sens. Environ., 236,
111493, https://doi.org/10.1016/j.rse.2019.111493, 2020.
Lizundia-Loiola, J., Franquesa, M., Khairoun, A., and Chuvieco, E.: Global burned area mapping from
Sentinel-3 Synergy and VIIRS active fires, Remote Sens. Environ., 282, 113298,
https://doi.org/10.1016/j.rse.2022.113298, 2022.
MacQueen, J.: Some methods for classification and analysis of multivariate observations, in:
Proceedings of the Fifth Berkeley Symposium on Mathematical Statistics and Probability, Volume 1:
Statistics, 281–298, 1967.
Mansuy, N., Boulanger, Y., Terrier, A., Gauthier, S., Robitaille, A., and Bergeron, Y.: Spatial attributes
of fire regime in eastern Canada: influences of regional landscape physiography and climate, Landsc.



1051 Ecol., 29, 1157–1170, https://doi.org/10.1007/s10980-014-0049-4, 2014.

1052 Marengo, J., Martinez, R., Tapia, B., Allen, T., Basantes, R., Hernandez-Espinoza, K., Alvarado, L.,
1053 Baddour, O., Ransom, C., Silva, Á., Báez, J., Gomez, F., Costa, F., Avalos, G., Estella, J., and Kennedy,
1054 J.: State of the Climate in Latin America and the Caribbean 2021 (WMO-No. 1295), 2022.

1055 Meinshausen, N.: Quantile Regression Forests, J. Mach. Learn. Res., 7, 983–999, 2006.

1056 Meteorological Organization, Naumann, G., Podestá, G., Marengo, J. A., Luterbacher, J., Bavera, D.,
1057 Arias Muñoz, C., Barbosa, P., Cammalleri, C., Acosta Navarro, J. C., Cuartas, L. A., Jager, A. de,
1058 Escobar, C., Hidalgo, C., Mazzeschi, M., Leal de Moraes, O. L., Estrada, M. de, Maetens, W., Magni,
1059 D., Masante, D., Seluchi, M. E., Milagros Skansi, M. de los, Felice, M. de, Fioravanti, G., Giordano, L.,
1060 Hrast Essenfelder, A., Osman, M., Rossi, L., Spennemann, P., Spinoni, J., Toreti, A., and Vera, C.:
1061 Extreme and long-term drought in the La Plata Basin: event evolution and impact assessment until
1062 September 2022 : a joint report from EC JRC, CEMADEN, SISSA and WMO, Publications Office of
1063 the European Union, 2023.

1064 Morello, J. H. and Adámoli, J. M.: Las grandes unidades de vegetación y ambiente del Chaco argentino,
1065 1968.

1066 Moreno, M. V., Laurent, P., and Mouillot, F.: Global intercomparison of functional pyrodiversity from
1067 two satellite sensors, Int. J. Remote Sens., 42, 9523–9541,
1068 https://doi.org/10.1080/01431161.2021.1999529, 2021.

1069 Mouillot, F., Schultz, M. G., Yue, C., Cadule, P., Tansey, K., Ciais, P., and Chuvieco, E.: Ten years of
1070 global burned area products from spaceborne remote sensing—A review: Analysis of user needs and
1071 recommendations for future developments, Int. J. Appl. Earth Obs. Geoinformation, 26, 64–79,
1072 https://doi.org/10.1016/j.jag.2013.05.014, 2014.

1073 Muñoz-Sabater, J., Dutra, E., Agustí-Panareda, A., Albergel, C., Arduini, G., Balsamo, G., Boussetta,
1074 S., Choulga, M., Harrigan, S., Hersbach, H., Martens, B., Miralles, D. G., Piles, M., Rodríguez-
1075 Fernández, N. J., Zsoter, E., Buontempo, C., and Thépaut, J.-N.: ERA5-Land: a state-of-the-art global
1076 reanalysis dataset for land applications, Earth Syst. Sci. Data, 13, 4349–4383,
1077 https://doi.org/10.5194/essd-13-4349-2021, 2021.

1078 Musser, K.: Río de la Plata, Wikipedia, 2024.

1079 Naumann, G., Podesta, G., Marengo, J., Luterbacher, J., Bavera, D., Acosta, N. J., Arias-Muñoz, C.,
1080 Barbosa, P., Cammalleri, C., Cuartas, L. A., De, E. M., De, F. M., De, J. A., Escobar, C., Fioravanti, G.,
1081 Giordano, L., Hrast, E. A., Hidalgo, C., Leal, D. M. O. L., Maetens, W., Magni, D., Masante, D.,
1082 Mazzeschi, M., Osman, M., Rossi, L., Seluchi, M., De, L. M. S. M., Spennemann, P., Spinoni, J., Toreti,
1083 A., and Vera, C.: Extreme and long-term drought in the La Plata Basin: event evolution and impact
1084 assessment until September 2022, https://doi.org/10.2760/62557, 2023.

1085 Naval-Fernández, M. C., Elia, M., Giannico, V., Bellis, L. M., Bravo, S. J., and Argañaraz, J. P.: The
1086 Pyrogeography of the Gran Chaco's Dry Forest: A Comparison of Clustering Algorithms and the Scale
1087 of Analysis, Forests, 16, 1114, https://doi.org/10.3390/f16071114, 2025.

1088 Nori, J., Torres, R., Lescano, J. N., Cordier, J. M., Periago, M. E., and Baldo, D.: Protected areas and
1089 spatial conservation priorities for endemic vertebrates of the Gran Chaco, one of the most threatened
1090 ecoregions of the world, Divers. Distrib., 22, 1212–1219, https://doi.org/10.1111/ddi.12497, 2016.

1091 Oliveras Menor, I., Prat-Guitart, N., Spadoni, G. L., Hsu, A., Fernandes, P. M., Puig-Gironès, R., Ascoli,
1092 D., Bilbao, B. A., Bacciu, V., Brotons, L., Carmenta, R., de-Miguel, S., Gonçalves, L. G., Humphrey,
1093 G., Ibarnegaray, V., Jones, M. W., Machado, M. S., Millán, A., de Morais Falleiro, R., Mouillot, F.,



Pinto, C., Pons, P., Regos, A., Senra de Oliveira, M., Harrison, S. P., and Armenteras Pascual, D.:
Integrated fire management as an adaptation and mitigation strategy to altered fire regimes, Commun.
Earth Environ., 6, 202, https://doi.org/10.1038/s43247-025-02165-9, 2025.
Olson, D. M., Dinerstein, E., Wikramanayake, E. D., Burgess, N. D., Powell, G. V., Underwood, E. C.,
and Kassem, K. R.: Terrestrial Ecoregions of the World: A New Map of Life on EarthA new global map
of terrestrial ecoregions provides an innovative tool for conserving biodiversity, BioScience, 51, 933–
1100 938, 2001.

Oom, D., Silva, P. C., Bistinas, I., and Pereira, J. M. C.: Highlighting Biome-Specific Sensitivity of Fire
Size Distributions to Time-Gap Parameter Using a New Algorithm for Fire Event Individuation, Remote
Sens., 8, 663, https://doi.org/10.3390/rs8080663, 2016.
Paritsis, J., Landesmann, J. B., Kitzberger, T., Tiribelli, F., Sasal, Y., Quintero, C., and Nuñez, M. A.:
Pine plantations and invasion alter fuel structure and potential fire behavior in a Patagonian forest-steppe
ecotone, Forests, 9, 117, https://doi.org/10.3390/f9030117, 2018.
Pausas, J. G. and Bradstock, R. A.: Fire persistence traits of plants along a productivity and disturbance
gradient in mediterranean shrublands of south-east Australia, Glob. Ecol. Biogeogr., 16, 330–340,
https://doi.org/10.1111/j.1466-8238.2006.00283.x, 2007.
Pettinari, M. L., Lizundia-Loiola, J., and Chuvieco, E.: ESA CCI ECV fire disturbance: D4. 2.1 product
user guide—MODIS, version 1.1, 2021.
Pielou, E. C.: The measurement of diversity in different types of biological collections, J. Theor. Biol.,
13, 131–144, https://doi.org/10.1016/0022-5193(66)90013-0, 1966.
Povak, N. A., Hessburg, P. F., and Salter, R. B.: Evidence for scale-dependent topographic controls on
wildfire spread, Ecosphere, 9, e02443, https://doi.org/10.1002/ecs2.2443, 2018.
Redford, K. H., Taber, A., and Simonetti, J. A.: There is More to Biodiversity than the Tropical Rain
Forests, Conserv. Biol., 4, 328–330, 1990.
Ruffault, J. and Mouillot, F.: How a new fire-suppression policy can abruptly reshape the fire-weather
relationship, Ecosphere, 6, art199, https://doi.org/10.1890/ES15-00182.1, 2015.
Ruffault, J. and Mouillot, F.: Contribution of human and biophysical factors to the spatial distribution
of forest fire ignitions and large wildfires in a French Mediterranean region, Int. J. Wildland Fire, 26,
498–508, https://doi.org/10.1071/WF16181, 2017.
Ruffault, J., Moron, V., Trigo, R. M., and Curt, T.: Objective identification of multiple large fire
climatologies: an application to a Mediterranean ecosystem, Environ. Res. Lett., 11, 075006,
https://doi.org/10.1088/1748-9326/11/7/075006, 2016.
Ruffault, J., Curt, T., Moron, V., Trigo, R. M., Mouillot, F., Koutsias, N., Pimont, F., Martin-StPaul, N.,
Barbero, R., Dupuy, J.-L., Russo, A., and Belhadj-Khedher, C.: Increased likelihood of heat-induced
large wildfires in the Mediterranean Basin, Sci. Rep., 10, 13790, https://doi.org/10.1038/s41598-020-
70069-z, 2020.
San Martín, R.: Fires, land use, and forest loss in the South American Chaco : understanding the links
between fires, climate, ecosystems, and human activity through remote sensing, PhD Thesis, Université
Paris-Saclay, 2024.
San Martín, R., Ottlé, C., and Sörensson, A.: Fires in the South American Chaco, from dry forests to
wetlands: response to climate depends on land cover, Fire Ecol., 19, 57, https://doi.org/10.1186/s42408-
1135 023-00212-4, 2023.



Saucedo, G. I. and Kurtz, D. B.: Seasonality and post fire recovery in a wetland dominated region:
Insights from satellite data analysis in northern Argentina, Remote Sens. Appl. Soc. Environ., 37,
101480, https://doi.org/10.1016/j.rsase.2025.101480, 2025.
Shannon, C. E.: A Mathematical Theory of Communication, Bell Syst. Tech. J., 27, 379–423,
https://doi.org/10.1002/j.1538-7305.1948.tb01338.x, 1948.
Sugiyama, M. S., Mendoza, M., and Carpio, M. B.: Resilience and Recovery in the Dry Chaco:
Ecological Knowledge Encoded in Forager Wildfire Narratives, J. Ethnobiol., 45, 76–94,
https://doi.org/10.1177/02780771241303896, 2025.
Takacs, S., Schulte to Bühne, H., and Pettorelli, N.: What shapes fire size and spread in African
savannahs?, Remote Sens. Ecol. Conserv., 7, 610–620, https://doi.org/10.1002/rse2.212, 2021.
Torrella, S. A. and Adámoli, J.: Situación ambiental de la ecorregión del Chaco Seco, Situac. Ambient.
Argent., 75–82, 2005.
Van Wagner, C. E.: Development and structure of the Canadian Forest Fire Weather Index System,
Minister of Supply and Services Canada, Ottawa, 37 pp., 1987.
Vidal-Riveros, C., Souza-Alonso, P., Bravo, S., Laino, R., and Ngo Bieng, M. A.: A review of wildfires
effects across the Gran Chaco region, For. Ecol. Manag., 549, 121432,
https://doi.org/10.1016/j.foreco.2023.121432, 2023.
Vidal-Riveros, C., Watler Reyes, W. J., Ngo Bieng, M. A., and Souza-Alonso, P.: Assessing Fire
Regimes in the Paraguayan Chaco: Implications for Ecological and Fire Management, Fire, 7, 347,
https://doi.org/10.3390/fire7100347, 2024.
Vitolo, C., Di Giuseppe, F., Barnard, C., Coughlan, R., San-Miguel-Ayanz, J., Libertá, G., and
Krzeminski, B.: ERA5-based global meteorological wildfire danger maps, Sci. Data, 7, 216,
https://doi.org/10.1038/s41597-020-0554-z, 2020.
Wright, M. N. and Ziegler, A.: ranger: A Fast Implementation of Random Forests for High Dimensional
Data in C++ and R, J. Stat. Softw., 77, 1–17, https://doi.org/10.18637/jss.v077.i01, 2017.
Yebra, M., Scortechini, G., Badi, A., Beget, M. E., Boer, M. M., Bradstock, R., Chuvieco, E., Danson,
F. M., Dennison, P., Resco de Dios, V., Di Bella, C. M., Forsyth, G., Frost, P., Garcia, M., Hamdi, A.,
He, B., Jolly, M., Kraaij, T., Martín, M. P., Mouillot, F., Newnham, G., Nolan, R. H., Pellizzaro, G., Qi,
Y., Quan, X., Riaño, D., Roberts, D., Sow, M., and Ustin, S.: Globe-LFMC, a global plant water status
database for vegetation ecophysiology and wildfire applications, Sci. Data, 6, 155,
https://doi.org/10.1038/s41597-019-0164-9, 2019.
Zhang, Y., Mao, J., Ricciuto, D. M., Jin, M., Yu, Y., Shi, X., Wullschleger, S., Tang, R., and Liu, J.:
Global fire modelling and control attributions based on the ensemble machine learning and satellite
observations, Sci. Remote Sens., 7, 100088, https://doi.org/10.1016/j.srs.2023.100088, 2023.