# Peer review of "What controls fire size in the South American Gran Chaco?"

_EGUsphere, 2025_

## Referee Comment (RC1)

**What controls fire size in the South American Gran Chaco? Exploring atmospheric, landscape, and anthropogenic drivers.**

Research article is tackling an important question regarding the controls of fire size (burned area) in the two different parts (xeric and mesic) of the Gran Chaco. It is an important subject, focusing on a special and diverse region and ecosystem. In methodology, researchers use different approaches throughout their analyses, and the article is well written. However, the connection between the fires in the Gran Chaco, their impact, and the underlying human-ecosystem dynamics is poorly established. I recommend the acceptance of the manuscript following the below revisions.

*major revision.*

**General comments:**

Since fire, especially in complex ecosystems such as the Chaco, is a product of vegetation composition, and since the authors specified landscape as one of their focus areas, I expected to see more on the vegetation type in the region (not a general grouping of herbaceous, shrubland, etc, but taxa that have a fire history and/or adaptation, and how they differ between the different parts of the Gran Chaco), the interplay between vegetation/topography/fire. However, the article centers its focus on meteorological conditions leaving the human and landscape elements to the side. This, I find is quiet unfortunate and also a lost opportunity studying the fire behavior in this truly magnificent landscape from different perspectives. At this stage the title falls short of meeting the expectations. Also, the area being very special and diverse, a short history of the past human/landscape interactions resonating in its current architecture would be informative to the reader who is not familiar with the region (especially if human use of fire and how anthropogenic ignitions over time may have changed), and would fulfill the expectations raised by the title. However, the authors have meticulously attempted to resolve the meteorological parameters.

The use of FireCCI51 is unfortunate, and I urge the authors to make the absolutely necessary comparison with its updated version FireCCI511. Please see line specific comments below on this. Also, the statistical methods used should take into consideration the underlying assumptions and should be reconsidered/changed where the criteria do not meet the requirements.

The scale considered for megafires should be revisited, explained (as to why ~10,000ha is selected as opposed to a larger number (i.e. 40,000+ constituting a megafire) as different agencies have different definitions of what is a "megafire". And also, an overall assessment of the impact of these fires is missing, and would be beneficial.

We already know from previous published research that fire severity and burned area are influenced by present meteorological conditions, climate variability (anomalies in wet/dry periods), fuel build up, wildland-urban-interface etc. However, in the analyses where I find lengthy analyses of meteorological conditions, i.e. the human influence is either weakly suggested or overall missing. Again, the researchers may wish to change the title and modify the abstract. Leaving out the anthropogenic element in such a region with such pronounced (and changing human impact) would be unfortunate however, and would certainly negatively affect the novelty of the research. Considering the largest driver of deforestation problem in the

region is anthropogenic, and there is a clear human impact on the fires in the Chaco, I expect a better-established discussion and analyses of the human drivers. In section 4.5, a weak reference to the human element is suggested but this falls short of a literature review. Also, the only reference to indigenous fire practices (or agriculture for that matter) is in the introduction, and the authors seem to have left it at that, and that is rather unfortunate. Additionally, and similarly, the landscape/vegetation connection is also very brief and weak. Again, may be a change in the title, with removal of these suggestions and a stronger connection to climate variability and anomalous fuel build up would better guide the reader and keep the expectations in check.

Finally, presentation of the results needs more structuring and if the authors wish to continue with the meteorology-vegetation-human narrative, then more detail on the later two drivers. Currently, the manuscript reads as a detailed analyses of the meteorological conditions, with a very generalized brush through the other two potential drivers. The authors train RF models, and even the results put land cover and vegetation in higher priority than wind and precipitation. I strongly suggest this to be taken into consideration.

In section **4.6** the authors list this under limitations, but if so, the study design should have been changed (along with the title). However the authors have not fully made use of the datasets which they have access to, to analyze the crossovers between wildfire and human presence.

Specific/detailed comments are listed below.

**Line specific notes/corrections:**

**70** indigenous (not capitalized)

**73** "…landscape context to assess how fire size responds to both short-term anomalies and long-term…" meteorogical extremes? Climate anomalies? Or something else?

**76** FireCCI51 is an older version, now deprecated. There is now FireCCI511cds available from Copernicus Data Store (cds). I recommend you consider using the updated version, or compare both versions to see if there are important differences between the datasets for your study area: Fire burned area from 2001 to present derived from satellite observations. I understand you use FRYv2.0 which incorporates the older version. But still, it would be more sensible to incorporate the updated dataset, especially if there are important spatio-temporal differences, or compare both datasets to see if there are discrepancies which may affect your final results:

*During July 2020, an error in some files in the version v5.1cds were identified, affecting the files of the grid product of January 2018, and the pixel and grid products of October, November and December 2019. These errors were fixed, and a new version, v5.1.1cds, was created for the whole time series, to replace version v5.1cds. The latter product has been deprecated, but it is temporally kept in the database for transparency and traceability reasons. Only version v5.1.1cds should be used.*

**193** *Also see abstract*: your reference to mega and giga-fires: it would be best if you were more explicit as to by which definition you are considering a 100+ km2 fire (which under the 40,000 ha min. for a megafire is a only large fire at ~10,000 ha) a megafire. These specifications are important as different agencies may use different scales.

**226** What constitutes "very" small/large? Again, I prefer the scale is referenced properly. Alternatively, you may say smaller or larger events, since you are already giving a spatial window of 1-100 km2.

**Fig 3**. The x-axis legends are barely readable. Also, a clear peak can be seen in year 2020, I expect to read up about what happened on that year. And why the ignitions show a downward trend between 2003-2019, then a peak at 2020. A change in policies?

**373** So the peak in 2021 is explained here by fire-weather anomalies (which I still have not seen, a show of FWI95 would be nice) but is that all? Considering roughly >80% of wildfires on a global scale originate from human ignitions, tying an anomalous year to only fire weather skips on the larger part of the narrative, the anthropogenic influence in the region.

**398-399** You are using Pearson correlation here, the prerequisite for which is normal distribution, however you previously mentioned that BA is skewed, suggesting a Poisson distribution.

**413** I don't know what you mean by FWI "anomaly" (whether it is in the climatic sense, the difference from a long term mean) but why not consider FWI95, the extremes? This is often a better indicator of how FWI shapes up overtime. In either case it would be helpful if you explained what constitutes an anomaly in your analysis.

**537** You have trained RF models which prioritize vegetation and topography but in your results you prioritize meteorological factors, which do not offer much novelty (there are already several studies which show FWI peaks and BA relationships, wind and BA and severity). The teleconnections are barely touched upon with no real analyses of the climatic changes in the region (is dry Chaco drying faster than before? Is wet Chaco experiencing drought episodes?). I strongly suggest you build your case (and discuss it) more strongly than you already are. If you wish to build a case around fire meteorology, then I strongly suggest you raise expectations to that effect (in the title, abstract and the main text).

**620** I'm yet to see a sign of a digital elevation model showing me the different parts of the terrain. Also, since you mention you used SRTM 1km, you could have layered it in your map plots so the reader could acquaint themselves with the terrain, the BA frequency, etc. In this section you generalize the role of vegetation in fire spread in a couple of lines thrown in with landscape. Whereas your title sets the tone as if this section should discuss one of the three drivers you specify, it is short and over generalized. Were there any refugia in your terrain, created by ie. Changes in vegetation patterns, topography, soil, climate? Do pyro catchments dominate. If so have you identified any and why do those spots burn more frequently (or less frequently) and is this pattern changing, etc etc.

---

## Author Comment (AC1)

**Author Responses to Reviewer 1**

**General overview**

We thank the reviewer for this constructive feedback. Our work is fully based on remote-sensing analysis, which naturally limits the investigation of species-level vegetation or socio-anthropological aspects. The study aims to identify and quantify drivers of fire size, rather than reviewing fire ecology or analyzing drivers of fire occurrence, which would require interdisciplinary field, policy, and social data across three countries.

We acknowledge that the paper is meteorology-centered because weather largely determines when fires expand, while fuels and landscape structure control how far they spread. This complements rather than replaces ecological or human-focused studies on Chaco fires. We will revise the title and abstract to better reflect this scope.

**1. Vegetation, landscape, and human context**

**Reviewer comment:**

Expected more information on vegetation types, landscape heterogeneity, and historical human use of fire.

**Author response:**

We will expand the Introduction and Discussion to include brief contextual information on **vegetation structure**, **landscape diversity**, **and human fire use**, with targeted references (Bravo et al.; Vidal-Riveros et al. 2023; our previous works, including San Martin's PhD thesis). However, a detailed biological or historical reconstruction lies **beyond the scope** of a remote-sensing study. These limits will be explicitly clarified, with references to existing comprehensive reviews.

**2. FRY / FireCCI provenance**

**Reviewer comment:**

FireCCI51 is outdated; use or compare it with FireCCIv.5.1.1cds. Only v5.1.1cds should be used.

**Author response:**

Our analysis uses **FRY v2.0**, which was generated from **FireCCI51** distributed on the ESA CCI portal. After a minor processing issue affecting Jan 2018 and Oct–Dec 2019 was identified, ESA CCI updated FireCCI51 in place (same version label) while Copernicus CDS published an equivalent corrected stream as v5.1.1cds for traceability. Consequently, **FireCCI51 downloaded from ESA CCI's catalog is equivalent to FireCCI v5.1.1cds (Copernicus Data Store); FRY v2.0**

was computed from the corrected CCI dataset, so our work already relies on the current, corrected inputs.

This has been confirmed in writing by María Lucrecia Pettinari (ESA FireCCI team) and Florent Mouillot (FRY developer and co-author of this manuscript). M.L. Pettinari clarified (email, 15 Oct 2025) that CCI's FireCCI51 and CDS's FireCCI v5.1.1cds are the same content under different versioning policies, and that FireCCI51 used for FRY was already corrected. F. Mouillot advised citing FireCCI51 (ESA CCI) as the official product name and DOI. We will add a concise provenance note in Methods explaining this equivalence and avoiding future confusion between the CCI and CDS labels. For transparency, we have attached the paraphrased email confirmation at the end of this document.

ESA CCI Fire data: <a href="https://climate.esa.int/en/projects/fire/">https://climate.esa.int/en/projects/fire/</a>

FireCCI51 from ESA CCI's catalog:

https://catalogue.ceda.ac.uk/uuid/3628cb2fdba443588155e15dee8e5352/

**3. Fire-size classification**

**Reviewer comment:**

Define and justify megafire thresholds; agencies use different scales: "why  $\sim$ 10,000 ha is selected instead of a larger number (e.g., 40,000+ ha)".

**Author response:**

Our classification of **Megafires** (100–1000 km² or 10,000–100,000 ha) and **Gigafires** (1000–10,000 km² or 100,000–1,000,000 ha) follows the standardized terminology proposed by Linley et al. (2022). For smaller fires, we used a classification adapted to the fire sizes we found in the Gran Chaco.

Linley et al. (2022) established globally comparable definitions for large-fire categories to promote clarity across studies and avoid inconsistencies with regional or agency-specific thresholds such as 40,000 ha.

As stated in Linley et al. (2022):

"To overcome ambiguity, we suggest a definition of megafire as fires > 10,000 ha arising from single or multiple related ignition events. We introduce two additional terms — gigafire (>  $100\,000$  ha) and terafire (>  $1\,000\,000$  ha) — for fires of an even larger scale."

This standardized framework ensures terminological consistency and facilitates direct comparison with global fire-size research. We already specify this classification and cite Linley

et al. (2022) in the Methods (line 194); in the revised version, we will make the reference more explicit and emphasize that our categories follow this scientifically defined scale.

**4. Human drivers and datasets**

**Reviewer comment:**

The anthropogenic dimension is weak; consider more analysis of human–fire interactions.

**Author response:**

We acknowledge this and will review Section 4.5 to improve clarity.

Besides, we recall that the changes made in the title of the paper will help readers not expecting a thorough analysis of human-fire interactions.

Additionally, we will **incorporate a more detailed road-network dataset to improve the calculation of road density within the fire polygons**, which is a clear and consistent proxy for accessibility and human influence measurable by remote sensing.

The dataset previously used likely under-represented informal or unmapped tracks, particularly in areas of private agricultural expansion or unregulated clearing.

**5. Figure 3 and 2020/21 peak**

**Reviewer comment:**

Improve readability and explain the 2020–2021 peaks.

**Author response:**

We will **rebuild Figure 3** (fonts, labels, scaling).

We may briefly explore additional potential explanations for the **2020 fire peak; however, our focus remains on fire size, rather than** fire occurrence or ignition causes. The years 2020 and 2021 were special from a sociological perspective, given the COVID-19 outbreak. It is possible to link some of the ignitions to a greater lack of governmental control, and the final sizes to a lack of suppression. Still, we believe that, given the strong La Niña event and the unprecedented drought that affected the region during that period, the causes of the **larger** fires were primarily meteorological.

**6. Pearson correlation and FWI anomalies**

**Reviewer comment:**

Clarify the use of Pearson correlation with skewed data and the meaning of FWI anomalies; why not use FWI95?

**Author response:**

We appreciate this observation and clarify that our correlations are performed on monthly anomalies of FWI and burned area (time series shown in Fig. A7 of the manuscript), not on the raw values. Only pixels with more than three fire-active months (burned area > 0) are included; those with fewer events or non-significant correlations are represented by dots in the maps and excluded from the statistics printed over the figures. In our revised manuscript, we will modify the filter to four fire-active months to improve robustness and avoid including pixels with too skewed burned area anomalies.

FWI anomalies are computed as daily deviations from the 2001–2020 climatology and then aggregated monthly. This definition is already described in the Methods (line 188), and we will make it clearer in the revised text. As shown in **Fig. 1 below**, the anomalies of burned area and fire counts are substantially less skewed than their raw values, supporting the validity of the correlation analysis.

**Fig. 1.** Histograms of FWI anomalies, Burned area (raw), Fire counts (raw), Burned area anomalies and fire counts anomalies for a particular pixel (coords: lat:  $-26.30^{\circ}$ ; lon:  $-56.7^{\circ}$ ).

Regarding the statistical method, Pearson's r does not strictly require normally distributed variables but assumes linearity and homoscedasticity. Because some burned area distributions

may remain right-skewed, we computed Spearman's rank correlations, which are non-parametric and robust to non-normality and outliers, following the reviewer's recommendation. We will include these updated figures in the revised manuscript.

The comparison (**Figures 2–3 below**) reveals that both approaches produce consistent spatial patterns across the Chaco: regions with positive correlations in the Wet and Dry seasons are similar in both cases. However, some differences in magnitude appear, particularly in transitional areas with weaker signal strength or more non-linear fire—weather responses (e.g., central Wet Chaco and parts of the Dry Chaco). These localized differences are expected because Spearman's  $\varrho$  downweights outliers and is less sensitive to extreme values than Pearson's r. Importantly, the persistence of the main spatial signal under both formulations demonstrates the robustness of our correlation patterns to methodological choice.

**Fig. 2.** Spatial correlation between monthly FWI anomalies and burned-area anomalies (2001–2022) computed using Pearson's r for the Wet and Dry seasons. Only pixels with at least four fire-active months and significant correlations (p

Fig. 3. Spatial correlation between monthly FWI anomalies and burned-area anomalies (2001–2022) computed using Spearman's r for the Wet and Dry seasons. Only pixels with at least four fire-active months and significant correlations (p < 0.05) are shown.

Finally, we emphasize that using pixel-level anomalies instead of a fixed FWI95 threshold allows each location to be evaluated relative to its own climatology, providing a more spatially consistent and locally meaningful measure of fire—weather relationships across the Chaco.

**7. DEM and topography**

**Reviewer comment:**

Add DEM and terrain context; discuss vegetation—topography—fire relations.

**Author response:**

A **DEM (SRTM)** is already included in **Figure 1**, as stated in the **Methods (lines 151–154)**: "Topography was derived from the Shuttle Radar Topography Mission (SRTM) digital elevation

model at 30 m resolution (https://srtm.csi.cgiar.org, accessed 26 May 2025) (...)"

We will make this clearer in the text, but we will not add the DEM to every man, as this work.

We will make this clearer in the text, but we will not add the DEM to every map, as this would obscure other layers.

**8. Minor edits and terminology**

**Reviewer comment:**

Various style and clarity issues.

**Author response:**

We will:

- Revise grammar errors and phrasing to "**short-term meteorological anomalies and long-term environmental gradients**";

**Summary of planned revisions**

- Clarify **FRY/FireCCI** provenance (with consultation of FRY's developer and co-author, Florent Mouillot, and FireCCI51 developer, Lucrecia Pettinari).
- Reaffirm and explain Linley fire-size classes (line 194).
- Add concise **vegetation**, **landscape**, **and human-use context** with references.
- Assess a **more detailed road dataset** to improve road-density estimation within fire polygons.
- Rebuild **Figure 3** for clarity.
- Clarify **FWI anomalies** (line 188) and justify **Pearson correlation**; add **Spearman** test.
- Emphasize that **pixel-level anomalies** are spatially consistent, unlike other metrics (e.g., FWI95).
- Clarify that **DEM** (SRTM) is already included in **Figure 1**.
- Revise **title/abstract** to better reflect the meteorological focus.

We thank the reviewer once again for their thoughtful and detailed evaluation. Their comments help us clarify the scope, strengthen methodological transparency, and improve the overall structure and readability of the manuscript.

---

## Author Comment (AC2)

**Author Responses to Reviewer 2**

**General overview**

We sincerely thank Reviewer 2 for their careful and constructive review. Your comments are very valuable in helping us identify sections that require clarification, deeper interpretation, and better integration with existing Chaco-specific literature. Below, we provide a structured response organized by topic.

**1. Interpretation and contextualization**

**Reviewer comment:** The manuscript would benefit from a deeper interpretation of results and stronger contextualization within the existing Chaco literature.

**Author response:** We fully agree. We will expand the discussion and interpretation of results by referring to additional studies that examine fire ecology, vegetation structure, and land-use dynamics across the Gran Chaco (e.g., Bravo et al. 2010; Argañaraz et al. 2015; Fischer et al. 2012; Naval Fernández et al. 2023). This will help strengthen the regional context and connect our findings to broader ecological and socio-environmental processes.

**2. Elevation and slope importance**

**Reviewer comment:** Elevation appears as the most important variable in the Random Forest models; the explanation is vague and should be explored further. Consider re-running the analysis without elevation.

**Author response:** We acknowledge this important comment. In a previous exploratory analysis not included in the manuscript, we had already tested Random Forest models excluding topographic variables (mean elevation and slope) to evaluate the relative importance of the remaining predictors. Those results showed that removing elevation and slope did not substantially alter the ranking of the other variables—the overall order of importance remained nearly identical to what can be visually inferred when disregarding elevation in the results currently presented.

In the revised manuscript, we will repeat the Random Forest analysis excluding elevation using the present dataset and covering all subregions and seasons, in order to formally assess its influence and confirm the robustness of our conclusions. Across all previous configurations, elevation consistently emerged as one of the most important predictors, which we interpret not as a direct causal factor but as a proxy for overlapping gradients in hydrology, vegetation structure, and land-use accessibility—all of which strongly influence final fire size.

We will expand the discussion to clarify this interpretation with concrete regional examples, supported by illustrative figures, such as the examples attached to this response, that could be added as supplementary material.

Some of the different direct and indirect effects of elevation can be summarized with the following examples:

**1. Wet Chaco - lowlands and islets (Fig. 1):**

In the floodplains of the Wet Chaco, elevation differences of only 10–20 m separate seasonally flooded lowlands from slightly higher islets ("albardones" or "montes" in Spanish) that support woody vegetation. Most fires occur in the lowlands, dominated by herbaceous cover that dries seasonally and sustains surface fires. We see that fires grow large as long as the lowlands are continuous, and that the 10-20 m higher islets act as barriers for fire growth. The tree-covered islets rarely burn, probably because trees form a discontinuous, moister fuel layer that interrupts the spread of low-intensity grass fires. Fires occurring over the islets or small lowlands patches surrounded by islets tend to be smaller than those occurring in large continuous lowlands. Elevation thus indirectly explains the spatial contrast in burned area by delineating zones with and without flammable vegetation.

**Fig. 1.** Topographic and land-cover context of a representative sector of the Wet Chaco. The upper left panel shows the Shuttle Radar Topography Mission (SRTM) digital elevation model at 90 m resolution. The upper right panel displays the same elevation surface overlaid with FRY v2.0 fire polygons (2001–2022) colored by fire-size class. The lower left panel presents a Google Earth high-resolution optical image of the area, and the lower right panel shows the ESA CCI Moderate-Resolution Land Cover (MRLC) map for 2022 (300 m).

**2. Wet Chaco - lowlands and islets (Fig. 2):**

In some other areas, height differences (usually around 20 m, but up to 60 m) coincide with human settlements established on elevated terrain next to rivers or floodplains—for example, the city of Asunción (Paraguay) or Corrientes (Argentina). These urbanized uplands show very low fire incidence. Although this reflects land use (fast human supression or lack of ignition) rather than topography per se, topography, which originally conditioned urban placement and accessibility, currently still serves as an indirect feature determining fire occurrence and particularly size.

**Fig. 2.** Topographic and land-cover context over the city of Asunción (Paraguay's capital city). The upper left panel shows the Shuttle Radar Topography Mission (SRTM) digital elevation model at 90 m resolution. The upper right panel displays the same elevation surface overlaid with FRY v2.0 fire polygons (2001–2022) colored by fire-size class. The lower left panel presents a Google Earth high-resolution optical image of the area, and the lower right panel shows the ESA CCI Moderate-Resolution Land Cover (MRLC) map for 2022 (300 m).

**3. Very Dry Chaco (mountainous Córdoba region) (Fig. 3):**

In contrast, the relationship reverses in the "Sierras". Elevated zones host dense, flammable shrublands with limited accessibility, where large wildfires dominate. The adjacent lowlands are used for agriculture and pastoral activities, where fires are generally smaller and more frequently managed or controlled. Here, slope and elevation enhance fire size through their association with vegetation type and suppression difficulty. Additionally, elevation here is related to orographic precipitation and a very marked difference in humidity on one or the other side of the mountain range running North-South. This difference affects vegetation types on either side of the

mountains, thus making topography an indirect driver not only of human presence and activity, but also of vegetation type and soil moisture.

**Fig. 3.** Topographic and land-cover context over the "Sierras de Córdoba" in the Very Dry Chaco (Argentina). The upper left panel shows the Shuttle Radar Topography Mission (SRTM) digital elevation model at 90 m resolution. The upper right panel displays the same elevation surface overlaid with FRY v2.0 fire polygons (2001–2022) colored by fire-size class. The lower left panel presents a Google Earth high-resolution optical image of the area, and the lower right panel shows the ESA CCI Moderate-Resolution Land Cover (MRLC) map for 2022 (300 m).

**4. Dry Chaco deforestation and agricultural fires (central Dry Chaco) (Fig. 4):**

In the gently undulating forested plains, elevation seems to play a minor role. Fires here mostly occur in deforested agricultural areas where fuel continuity is governed by land use rather than topography. The very gradual east—west elevation gradient has a less important ecological meaning for fire spread in this subregion.

**Fig. 4.** Topographic and land-cover context over a deforested area within the Gran Chaco forest in Argentina. The upper left panel shows the Shuttle Radar Topography Mission (SRTM) digital elevation model at 90 m resolution. The upper right panel displays the same elevation surface overlaid with FRY v2.0 fire polygons (2001–2022) colored by fire-size class. The lower left panel presents a Google Earth high-resolution optical image of the area, and the lower right panel shows the ESA CCI Moderate-Resolution Land Cover (MRLC) map for 2022 (300 m).

Taken together, these examples show that elevation functions as an integrative variable capturing multiple mechanisms—hydrological, ecological, and anthropogenic—that differ among Chaco subregions. Its high importance in the Random Forest models reflects its ability to summarize several latent gradients rather than a direct physical control on fire spread.

Additionally, to better represent human accessibility, we will incorporate a more detailed road-network dataset to refine the calculation of road density within fire polygons. This improvement will provide a stronger proxy for human influence and

allow us to test whether the prominence of elevation partly results from the current limitations of anthropogenic indicators.

Overall, these new analyses and supporting figures will clarify why elevation and slope appear as key predictors of fire size in our models and will directly address the reviewer's request for a deeper interpretation of their ecological meaning.

**3. Methodological details and performance metrics**

**Reviewer comment:** The paper should better connect with local literature and clarify methodological details, such as Random Forest performance metrics.

**Author response:** We will strengthen the discussion with key Chaco-specific studies and add model-performance metrics (e.g., out-of-bag R2, RMSE, which were inadvertently excluded) in the Results section to reinforce our results. The Methods will also clarify variable definitions, collinearity checks, and consistent terminology (e.g., using "fire counts" instead of "ignitions").

**4. Clarity and citations (L42–69)**

**Reviewer comment:** Revise the link between exotic grasses and fire intensity; ensure cited studies support the statements.

**Author response:** We will refine several sentences to ensure each claim is directly supported by cited literature. In particular, we will clarify the discussion on exotic grasses and their influence on fire intensity. We will note that such invaded areas are spatially limited within the Chaco and adjust the phrasing to avoid overgeneralization.

**5. Terminology and consistency (L192 ff.)**

**Reviewer comment:** Standardize terminology throughout the manuscript.

**Author response:** We will standardize terminology (e.g., "fire patches" instead of "fire polygons," "meteorological" instead of "climatic," and "fire counts" instead of "ignitions") to maintain coherence throughout the text.

**6. Data sources and resolution (L155–158)**

**Reviewer comment:** Explain why ESA CCI Land Cover was used instead of MapBiomas Chaco.

Author response: We selected ESA CCI Land Cover (300 m) because it provides a consistent, globally validated, and temporally continuous dataset directly compatible with FireCCI51 and FRY products. This ensures coherence between the fire and land-cover components of our analysis. MapBiomas Chaco dataset, although valuable, still shows some regional classification uncertainties and requires non-trivial extraction procedures through Google Earth Engine, which complicates reproducibility and temporal analysis. For our multi-year, regional-scale work, ESA CCI LC offers a robust and harmonized framework. This justification will be made explicit in the Methods section.

**7. Figures and tables**

**Reviewer comment:** *Improve readability of figures and captions.*

**Author response:** Figure and table captions will be expanded for clarity, color schemes harmonized, and all morphology indices explicitly defined in the Methods. We will also avoid opening sections with figures or tables.

**8. Fire Weather Types and morphology (L492 ff.)**

**Reviewer comment:** Clarification about FWTs and fire morphology.

**Author response:** We will expand the interpretation of how Fire Weather Types (FWTs) relate to fire morphology. Wind-driven fires tend to display higher elongation and compactness. This will be clearly discussed in the Results and Discussion.

**9. Discussion enrichment and literature integration (L583–669)**

**Reviewer comment:** Strengthen the Discussion by connecting with additional Chaco-specific research.

**Author response:** We will discuss our results in light of the regional studies suggested, some of which were already referenced in the introduction and some that will be added (e.g., Bravo et al. 2010; Argañaraz et al. 2015, 2016, 2018; Bianchi et al. 2014; Fischer et al. 2012). We will also correct the section on prescribed burning.

**10. Figures 10, 11, and 15**

**Reviewer comment:** *Improve interpretability and consistency across figures.*

**Author response:** We will enhance visual clarity (consistent color schemes, transparency for overlapping points). Figure 15 will explicitly describe the elevation and slope ranges of the selected data used for the Random Forests.

**Summary of planned revisions**

- Improved figure readability and consistent design.
- Clear justification for using ESA CCI Land Cover (300 m).
- Revise the introduction and discussion sections with recommended references
- Clarified discussion of informal versus prescribed burning in the Chaco.
- Revised Random Forest analysis excluding elevation, with added model-performance metrics. Improve the discussion of
- Evaluation of a more detailed road dataset to refine road-density estimation and check the robustness of topography as the main predictor for fire size.

We thank Reviewer 2 once again for the detailed and constructive comments, which will help to clarify our scope, improve methodological transparency, and strengthen our results.